# *Wilms Tumor 1b* defines a wound-specific sheath cell subpopulation associated with notochord repair

**Juan Carlos Lopez-Baez**[1,2], **Daniel J Simpson**[1†], **Laura LLeras Forero**[3,4,5†], **Zhiqiang Zeng**[1,2], **Hannah Brunsdon**[1,2], **Angela Salzano**[1], **Alessandro Brombin**[1,2], **Cameron Wyatt**[1], **Witold Rybski**[1,2], **Leonie F A Huitema**[3], **Rodney M Dale**[6], **Koichi Kawakami**[7], **Christoph Englert**[8,9], **Tamir Chandra**[1], **Stefan Schulte-Merker**[3,4,5], **Nicholas D Hastie**[1*], **E Elizabeth Patton**[1,2*]

[1]MRC Human Genetics Unit, MRC Institute of Genetics and Molecular Medicine, University of Edinburgh, Edinburgh, United Kingdom; [2]CRUK Edinburgh Centre, MRC Institute of Genetics and Molecular Medicine, University of Edinburgh, Edinburgh, United Kingdom; [3]Hubrecht Institute - KNAW & UMC Utrecht, Utrecht, Netherlands; [4]Faculty of Medicine, Institute for Cardiovascular Organogenesis and Regeneration, WWU Münster, Münster, Germany; [5]CiM Cluster of Excellence, Münster, Germany; [6]Department of Biology, Loyola University Chicago, Chicago, United States; [7]Division of Molecular and Developmental Biology, National Institute of Genetics, Mishima, Japan; [8]Department of Molecular Genetics, Leibniz Institute for Age Research-Fritz Lipmann Institute, Jena, Germany; [9]Institute of Biochemistry and Biophysics, Friedrich-Schiller-University, Jena, Germany

**\*For correspondence:**
Nick.Hastie@igmm.ed.ac.uk (NDH);
e.patton@igmm.ed.ac.uk (EEP)

†These authors contributed equally to this work

**Competing interests:** The authors declare that no competing interests exist.

**Abstract** Regenerative therapy for degenerative spine disorders requires the identification of cells that can slow down and possibly reverse degenerative processes. Here, we identify an unanticipated wound-specific notochord sheath cell subpopulation that expresses Wilms Tumor (WT) 1b following injury in zebrafish. We show that localized damage leads to Wt1b expression in sheath cells, and that *wt1b*[+]cells migrate into the wound to form a stopper-like structure, likely to maintain structural integrity. *Wt1b*[+]sheath cells are distinct in expressing cartilage and vacuolar genes, and in repressing a Wt1b-p53 transcriptional programme. At the wound, *wt1b*[+]and *entpd5*[+] cells constitute separate, tightly-associated subpopulations. Surprisingly, *wt1b* expression at the site of injury is maintained even into adult stages in developing vertebrae, which form in an untypical manner via a cartilage intermediate. Given that notochord cells are retained in adult intervertebral discs, the identification of novel subpopulations may have important implications for regenerative spine disorder treatments.
DOI: https://doi.org/10.7554/eLife.30657.001

## Introduction

Wilms tumour 1 (WT1) is a zinc finger transcription factor that regulates key developmental stages of several mesodermal tissues including the kidneys, gonads and coronary vasculature (*Hastie, 2017*). In the developing kidney, WT1 is required for the maintenance of mesenchymal nephron progenitors (*Kreidberg et al., 1993*; *Motamedi et al., 2014*) as well as differentiation of these progenitors into the epithelial components of the nephron (*Essafi et al., 2011*). In contrast, in the developing heart, WT1 is expressed in the epicardium (mesothelial lining) and required for the production, via an epithelial to mesenchymal transition (EMT), of coronary vascular progenitors (EPDCs) that migrate into

the myocardium (*Martínez-Estrada et al., 2010*). Similarly, WT1-expressing mesothelium is the source of mesenchymal progenitors for specialised cell types within several other developing organs. These include stellate cells within the liver (*Asahina et al., 2011*), interstitial cells of Cajal in the intestine (*Carmona et al., 2013*) and adipocytes within visceral fat depots (*Chau et al., 2014*). WT1 expression is down-regulated in the epicardium postnatally but reactivated in response to tissue damage in both mice (*Smart et al., 2011*) and zebrafish (*Schnabel et al., 2011*). In both organisms, this activation of WT1 in response to damage is associated with new rounds of epicardial EMT, leading to the production of coronary vascular progenitors (*Schnabel et al., 2011*; *Smart et al., 2011*).

Given the reactivation of *Wt1/wt1b* in the damaged epicardium we set out to investigate whether Wt1 programmes are initiated in response to other sources of tissue damage in zebrafish, and uncovered a novel Wt1 response to wounding of the notochord. The notochord is a transient embryonic structure that provides axial support and signalling information (*Stemple, 2005*). The notochord comprises two cell populations, the inner vacuolated cells that provide rigid support to the embryo, and the outer sheath cells, a single cell epithelial layer that surrounds the vacuolated cells and secretes components of the extracellular matrix to provide turgor pressure to the vacuolated cells (*Apschner et al., 2011*; *Ellis et al., 2013*). This rigid axial structure becomes functionally replaced by vertebra of the axial skeleton over time. In zebrafish, a row of metameric mineralized rings, known as chordacentra, forms around the notochord in an anterior to posterior fashion and constitutes the first signs of the definitive vertebral column. The chordacentra delineate the future sites where mature vertebra will form and ossify as the larva grows, while the notochord cells develop into the nucleus pulposus of the adult intervertebral disc, a soft gel-like tissue that provides cushioning and flexibility for the spine (*Parsons, 1977*).

Degeneration of the intervertebral disc leads to extensive back pain, one of the top global causes of years lived with disability (*Lawson and Harfe, 2015*). Treatment primarily consists of managing the pain symptoms, or in more progressed disease includes extensive surgery. One of the major goals of the tissue-engineering field is to identify cells and tissues that will enable novel regenerative therapies to slow down and possibly reverse the degenerative process. Here, we uncover a novel cellular subpopulation in the notochord sheath that emerges at the site of damage and is maintained until formation of a repaired adult vertebra structure. Surprisingly, this subpopulation expresses *wt1b* despite no evidence of *wt1b* expression in physiological notochord development or ossification. Our findings suggest that the zebrafish notochord is protected by a novel wound-specific programme that seals the notochord wound in the embryo and contributes to the subsequent adult vertebra at the injury site.

## Results

### Wound-specific expression of *wt1b* in the notochord

Given the expression of *wt1b* in the regenerating heart, we wanted to explore the expression of *wt1* in other regenerating tissues, and began with the tail fin regenerative processes. There are two *wt1* paralogues in zebrafish, *wt1a* and *wt1b*, and so we performed tail fin amputations on zebrafish larvae 3 days post fertilization (dpf) using *Tg(wt1a:gfp)* and *Tg(wt1b:gfp)* transgenic lines (*Bollig et al., 2009*; *Perner et al., 2007*) (*Figure 1—figure supplement 1a*). Surprisingly, we discovered that tail fin amputations including partial removal of the notochord triggered a change of cellularity in the notochord, coupled with the specific, de novo upregulation of GFP in a *Tg(wt1b:gfp)* transgenic line. This response was specific to *wt1b* because we did not observe expression of GFP in the notochord of *Tg(wt1a:gfp)* tail fin amputated larvae (*Figure 1—figure supplement 1b–f*).

Next, we developed a needle-based assay to induce localized damage in the developing zebrafish notochord independent of tail fin amputation. Needle injury was induced in 3 dpf *Tg(wt1b:gfp)* that had been crossed with *casper* fish to remove pigmentation and imaged at 72 hr post injury (hpi) (*Figure 1a*). Needle induced wounds triggered a similar, albeit stronger *wt1b:gfp* response compared to the tail fin amputations, that was specifically localised to the site of the wound (*Figure 1b*). Time course imaging showed a progressive expansion of the damaged area over 72 hr, with an increasing expression of GFP signal, concomitant with a change of cellularity in the notochord (*Figure 1c*). Importantly, this was not observed in uninjured zebrafish controls (*Figure 1c*) or in notochord injured *Tg(wt1a:gfp)* transgenic larvae (data not shown). Histological staining of the damaged

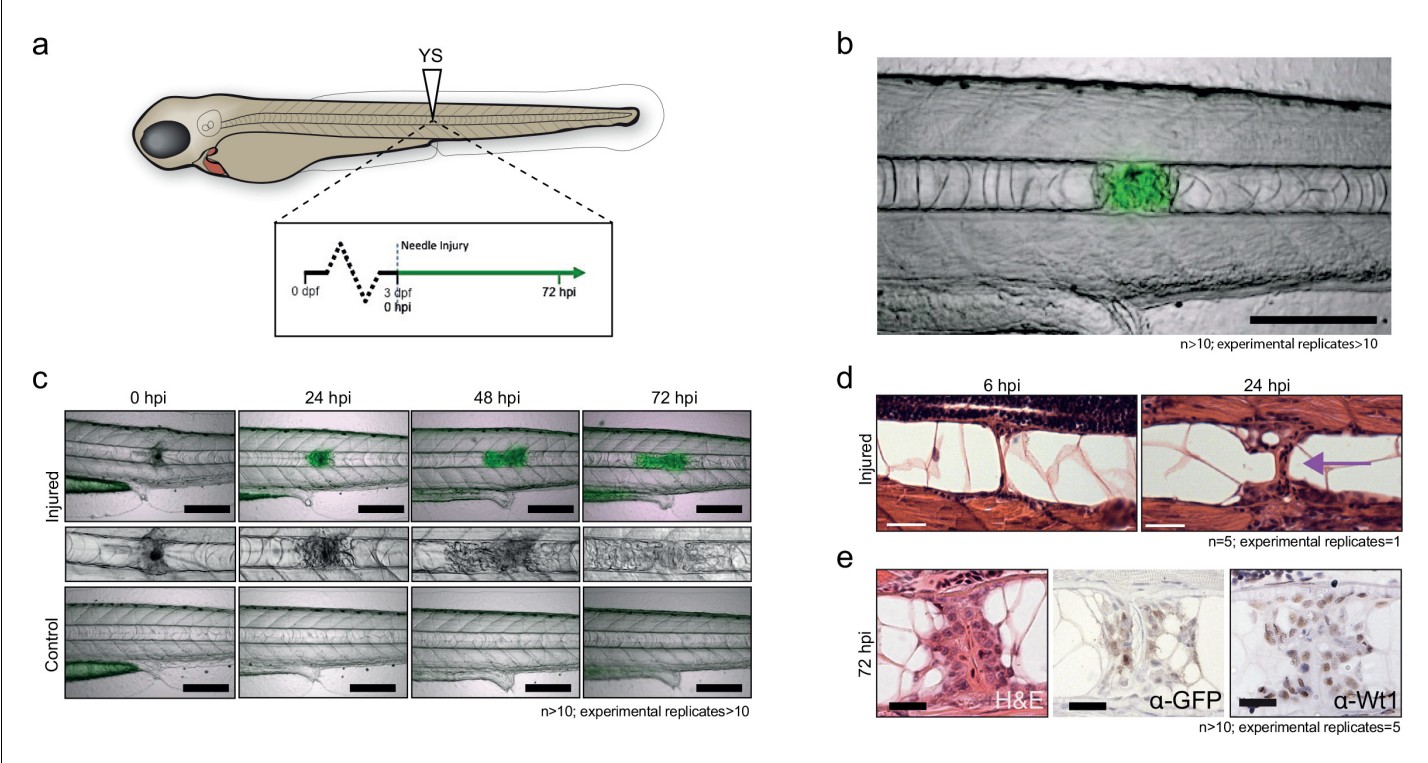

**Figure 1.** Notochord injury triggers local and sustained *wt1b* expression. (a) Schematic of notochord needle-injury protocol. 3 dpf *Tg(wt1b:gfp)*; casper larvae are injured above the yolk sac (YS; at somite 14 or 15) and followed for 72 hr. (b, c) Images of *Tg(wt1b:gfp); casper* zebrafish trunk over time following notochord needle injury, and uninjured matched controls. GFP signal is associated with a change of cellularity in the injured notochord (inset). n > 10; experimental replicates >10. Scale bar: 100 µm. (d) H and E staining of the injured area at 6 hpi and 24 hpi highlighted the progressive change in cellularity at the site of the injury (arrow). n = 5; experimental replicates = 1. Scale bar: 20 µm. (e) Immunohistochemistry of the injured area with α-GFP and α-Wt1 antibodies. n > 10; experimental replicates = 5. Scale bar: 20 µm. dpf = days post fertilization; hpi = hours post injury; H and E = haematoxylin and eosin.

DOI: https://doi.org/10.7554/eLife.30657.002

The following figure supplements are available for figure 1:

**Figure supplement 1.** *wt1b* expression in tail amputated larvae.
DOI: https://doi.org/10.7554/eLife.30657.003

**Figure supplement 2.** Wt1 and GFP protein expression in the notochord.
DOI: https://doi.org/10.7554/eLife.30657.004

area revealed the presence of a subpopulation of cells at the site of injury, which contrasted morphologically with the uniform, vacuolated inner cells of the notochord (*Figure 1d*). These cells stained positively for GFP and for endogenous Wt1 protein by immunohistochemistry, validating the faithful expression of the transgene with endogenous *wt1b* expression in this response (*Figure 1e*; *Figure 1—figure supplement 2*). *Tg(wt1b:gfp)* expression was not detected in the notochord outside the wound response by immunohistochemistry for GFP or for Wt1 protein (*Figure 1—figure supplement 2*). Thus, following notochord injury, an unanticipated expression of *wt1b* marks a subpopulation of cells that emerges in the notochord and is associated with the wound.

## *wt1b* expressing cells emerge from the notochord sheath

To determine the origin of the wound-specific *wt1b*⁺ cells, we examined *wt1b* expression in the vacuolated cells of the notochord, and in notochord sheath cells using two different transgenic lines. The *Tg(SAGFF214A:gfp)* transgenic line labels the cytoplasm of the inner vacuolated cells, and the *Tg(R2col2a1a:mCherry)* transgenic line labels notochord sheath cells. While *col2a1a* is expressed in all notochord cells (*Apschner et al., 2011*), a *Tg(R2col2a1a:mCherry)* line had been generated with a 310 bp conserved regulatory element of the *col2a1a* promoter that is specifically expressed in the

surrounding notochord sheath cells (*Figure 2a*) (*Dale and Topczewski, 2011*; *Yamamoto et al., 2010*).

A needle-induced notochord wound in the *Tg(SAGFF214A:gfp)* transgenic line showed that GFP-expressing cells were lost rapidly upon injury, creating a gap in the row of vacuolated cells. Eventually, this gap was filled with new cells by 144 hpi (*Figure 2—figure supplement 1a,b*). The *SAGFF214A:gfp* response was distinct from the *wt1b*[+] response in time (emerging at 72 hpi compared with 24 hpi), size and number (few and large compared with numerous and small), and in coverage of the wound (visible gaps remaining at the site compared with filling the damage site). These data suggest that *wt1b* expressing cells are distinct from the vacuolated cells at the site of injury.

Next, we explored the role of the notochord sheath cells in this process. We crossed the *Tg (wt1b:gfp)* transgenic line to the *Tg(R2col2a1a:mCherry)* transgenic line. Live confocal and multiphoton imaging revealed *wt1b:gfp* expression in the *R2col2a1a:mCherry* notochord sheath cells following needle induced notochord damage (*Figure 2b–d*; *Video 1*; *Figure 2—figure supplement 1c*), and this was supported by imaging of histological sections (*Figure 2—figure supplement 1d*). *wt1b:gfp* co-expression with *R2col2a1a:mCherry* was visible by 24 hpi in a ring surrounding the notochord vacuolated cells, and by 72 hpi the *wt1b:gfp* subpopulation of sheath cells had migrated into central aspects of the notochord to fill the wound and produce a visible stopper-like seal that was contiguous with the notochord sheath cells, and filled the gap in the notochord caused by the wound (*Figures 1e* and *2d*).

To validate the co-expression of *wt1b:gfp* and *col2a1a:mCherry* in the wounded fish, we FACS sorted cell populations in the injured versus uninjured larvae isolated from the trunk region (*Figure 2e*; 35 larvae pooled per set). Both injured and non-injured larvae contained cells that expressed either GFP[+] only (presumably *wt1b:gfp* cells of the pronephric duct that were included in the dissected tissue) or mCherry[+] alone, but the wounded fish had significantly increased numbers of cells that co-expressed *wt1b:gfp* and *col2a1a:mCherry* (GFP[+]mCherry[+]) (*Figure 2—figure supplement 1e*).

Our evidence indicates that the notochord wound triggers a unique *wt1b*[+] subpopulation to emerge in the notochord sheath cells. This *wt1b*[+] sheath cell subpopulation migrates into the wound and generates a stopper-like structure, possibly to prevent further loss of notochord turgor pressure and maintain notochord integrity.

## Nystatin mediated disruption of vacuolated cells leads to an increase in *wt1b:gfp* expression

We tested if the *wt1b*-response was specific to wounds that involved rupture of the sheath, or if *wt1b* expressing cells could be induced upon loss of vacuolated cell integrity alone. Mutations in *caveolin* genes lead to collapse of the vacuolated cells, with invasion and replacement from the notochord sheath (*Garcia et al., 2017*). We treated two-day old *Tg(wt1b:gfp; R2col2a1a: mCherry)* zebrafish with nystatin, a small molecule that binds sterols. Nystatin treatment lead to an increase in cellularity of the vacuolated notochord, similar to the phenotype seen in the notochord of *caveolin* mutants (*Figure 2—figure supplement 2*). GFP was expressed in a subpopulation of the mCherry-positive sheath cells at the site of cellularity. Thus, expression of *wt1b* in the sheath does not require a physical breach of the sheath, and *wt1b* expression may be applicable to a wider range of tissue stress and damage situations.

## Notochord wound cells express cartilage and mesenchyme genes

To address the molecular process at the site of the wound, we compared the transcriptome of

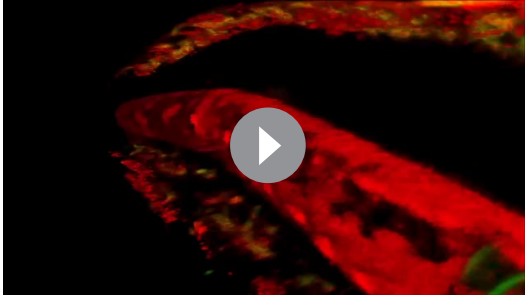

**Video 1.** Time-lapse imaging of two-photon microscopy of *Tg (wt1b:gfp; R2col2a1a:mCherry)* zebrafish larvae following needle injury over 48 hr. *wt1b:gfp* expression is upregulated in *R2col2a1a: mCherry* expressing notochord sheath cells upon needle injury, leading to the formation of a stopper like structure across the wound
DOI: https://doi.org/10.7554/eLife.30657.009

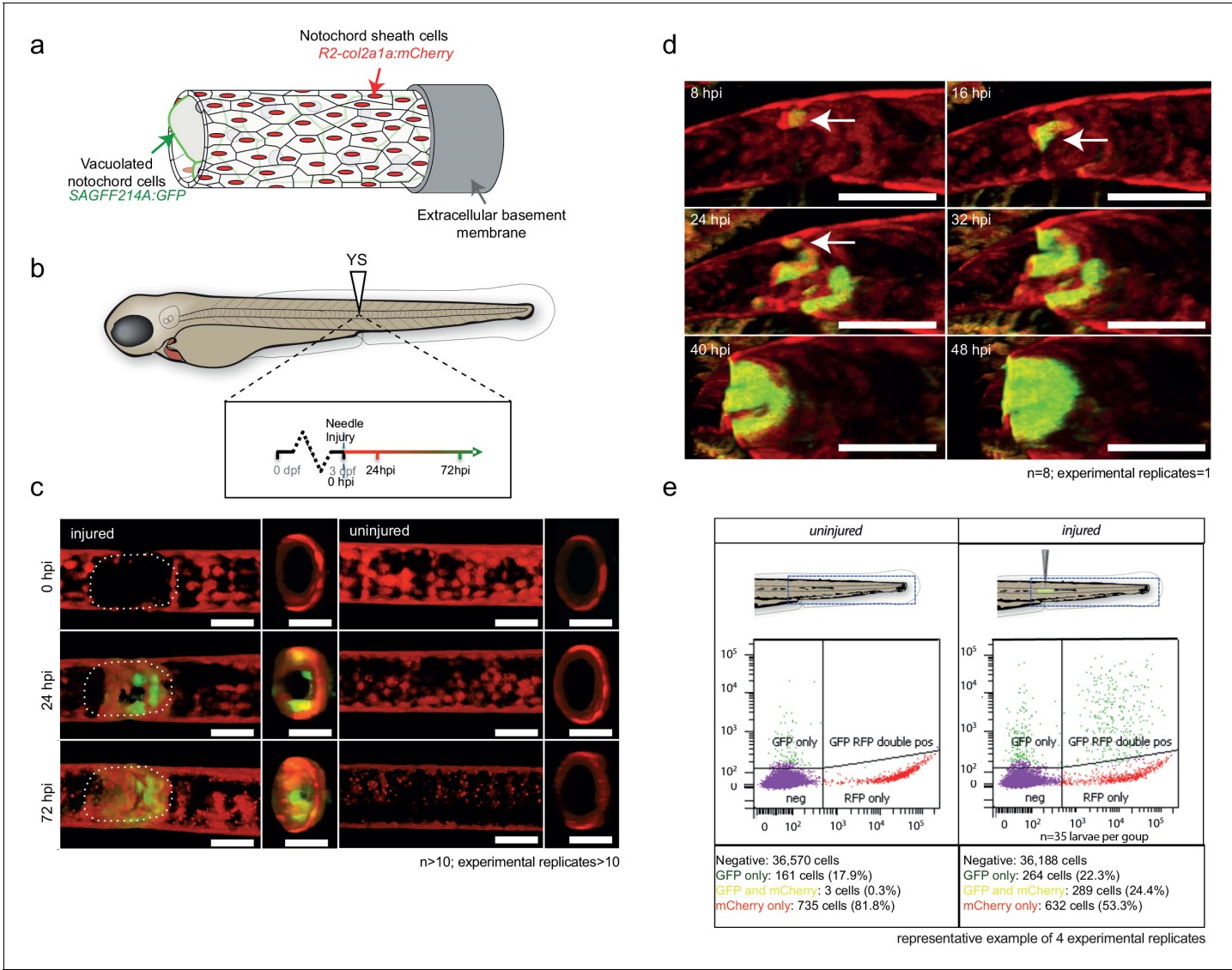

**Figure 2.** *wt1b:gfp* expressing notochord sheath cells populate the site of injury in the damaged notochords. (a) Schematic diagram of the notochord and transgenic lines used in this study. The notochord is composed of an inner population of highly vacuolated cells (green arrow; *SAGFF214A:gfp*), surrounded by a layer of epithelial-like sheath cells (red arrow; *R2col2a1a:mCherry*), encapsulated by a thick layer of extracellular basement membrane (grey arrow). (b) Schematic of experimental design: 3dpf *Tg(wt1b:gfp; R2-col2a1a:mCherry); casper* larvae were needle-injured and imaged at 0, 24 and 72 hpi. (c) Needle damage led to the formation of a cell-free gap in the layer of notochord sheath cells (0 hpi – injured; dashed line). GFP expression can be observed in the notochord sheath cells surrounding the area of damage by 24 hpi (inset: cross-sectional view) and these appear to engulf the injured area by 72 hpi (inset). n > 10; experimental replicates >10. Scale bar: 100 μm. (d) Multiphoton time-lapse imaging of wound site. Initial upregulation of GFP occurs at eight hpi in the *R2-col2a1a:mCherry* positive cells (arrow) and propagates across the injured area over the next 40 hr to form a seal in the notochord. n = 8; experimental replicates = 1. Scale bar: 100 μm. (e) Representative example of FACS analysis of cell populations in injured and non-injured zebrafish trunk tissue. GFP$^+$mCherry$^+$ double positive cells are present in injured *Tg(wt1b:gfp; col2a1a:mCherry)* at 72 hpi. Percentage of fluorescent cells are reported. Note that the dissected tissue can also encompass *wt1b:gfp* expressing cells in the posterior end of the pronephric duct (see also *Figure 1c*). n = 35 larvae per group; experimental replicates = 4. dpf = days post fertilization; hpi = hours post injury.

DOI: https://doi.org/10.7554/eLife.30657.005

The following source data and figure supplements are available for figure 2:

**Figure supplement 1.** Imaging cell populations at the wound.

DOI: https://doi.org/10.7554/eLife.30657.006

**Figure supplement 1—source data 1.** Raw data and statistical analyses for *Figure 2—figure supplement 1e*.

DOI: https://doi.org/10.7554/eLife.30657.008

**Figure supplement 2.** Nystatin treatment leads to upregulation of *wt1b:gfp* expression in notochord sheath cells.

DOI: https://doi.org/10.7554/eLife.30657.007

the trunk region in the injured and uninjured 72 hpi larvae (*Figure 3a,b*; n = 50 larvae per subset). Microarray analysis revealed a highly significant 131-fold increase in expression of *matrix gla protein* (*mgp*), a gene that is known to express in chondrocytic zebrafish tissues (*Gavaia et al., 2006*) and to be involved in the inhibition of hydroxyapatite production during ectopic bone formation (*Schurgers et al., 2013*; *Sweatt et al., 2003*; *Zebboudj et al., 2002*) (*Figure 3c,d*). Other genes included mesenchymal and cell adhesion markers, such as *fn1b*, coagulation factors, such as *f13a1b*, and immune response genes, such as *zgc:92041* and *complement c6* (*Figure 3d*).

The increased expression of *mgp* and *f13a1b* genes implicated the de novo acquisition of chondrogenic features in the injured tissues. Chondrogenic cells in the endochondral tissues of the craniofacial, fin bud and axial skeletons express *mgp* (*Gavaia et al., 2006*) and *FXIIIA* expression is localized to the developing chondrogenic mesenchyme of the pectoral fin bud (*Deasey et al., 2012*). The expression of cartilage genes was unexpected because ossification around the zebrafish notochord occurs via the formation the chordacentra, and does not require the establishment of cartilage anlagen (*Bensimon-Brito et al., 2012*; *Fleming et al., 2004)*. To examine the expression of other chondrogenic genes, we analyzed the top 100 significant genes and found an increase in expression of *sox9b*, the master regulator of chondrogenesis, five collagen genes associated with chondrogenic tissues (*col2a1a*, *col2a1b*, *col11a2*, *col9a1* and *col9a2*), the cartilage-specific extracellular structural protein Aggrecan, a microRNA regulator of chondrogenesis microRNA140 and the matrix-cell anchor protein chondroadherin (*chad*) (*Figure 3e*). To validate these findings at the molecular level, we isolated sections of damaged and undamaged tissue, and performed qRT-PCR for *matrix gla protein* (*mgp)* and *sox9b*. We chose these two genes because *mgp* was highly expressed in the microarray analysis and important for bone organization, and because Sox9 is a master cartilage transcription factor. We found *mgp* and *sox9b* to be highly upregulated in the injured tissue compared with the uninjured tissue (*Figure 3f,g*). Our results reveal that notochord wounding leads to the formation of a *wt1b*-positive sheath subpopulation that is characterised by an unexpected increase in genes associated with cartilage.

## Single-cell and 10 cell sequencing of *wt1b*-expressing sheath cells

To address the molecular nature of the GFP$^+$mCherry$^+$ expressing cells, we performed RNA sequencing of single-cells and 10 cell pools of FACS sorted GFP$^+$ cells, mCherry$^+$ cells and GFP$^+$mCherry$^+$ cells from injured zebrafish (3dpi) using the SMARTseq2 protocol (*Supplementary file 1*; *Figure 4—figure supplement 1*) (*Kirschner et al., 2017*; *Picelli et al., 2013*). To avoid batch effects, all experimental conditions were sorted onto the same 96 well plate and processed simultaneously (*Baran-Gale et al., 2017*). Sequencing reads were processed using the Scater pipeline (*McCarthy et al., 2017*). Unbiased Single cell consensus clustering (SC3) of the whole transcriptomes revealed that the GFP$^+$ cells, mCherry$^+$ cells and GFP$^+$mCherry$^+$ cells clustered into three distinct subpopulations (SC3 cluster 1: GFP$^+$, 2: GFP$^+$mCherry$^+$ and 3: mCherry$^+$) (*Figure 4a–c*) (*Kiselev et al., 2017*). Single and 10 cell populations clustering together suggested that sorting conditions led to homogenous 10 cell populations. Expression of *wt1b* was detected in SC3 clusters 1 and 2, and *col2a1a* was expressed in SC3 clusters 2 and 3 (*Figure 4b*). *wt1a* transcripts were not detected in any of the SC3 clusters. Together with the Wt1b antibody immunohistochemistry (*Figure 1e*, *Figure 1—figure supplement 2*), detection of *wt1b* transcripts in GFP$^+$mCherry$^+$ cells prove endogenous *wt1b* expression in the notochord damage response.

To avoid confounding factors, for example different ratios of single to 10 cell trancriptomes, when calculating differential expression, we used SC3 on the 10 cell populations only. We found consistent clustering of the different cell populations (GFP$^+$, GFP$^+$mCherry$^+$ and mCherry$^+$). Notably, differential marker gene expression in GFP$^+$mCherry$^+$ cells included the *mgp*, *fn1b* and *f13a1b* genes (*Figure 4c*) that were highly upregulated in the wounded tissue (*Figure 3d*). To validate our findings, we isolated injured notochord tissue from 3dpi and FACS sorted GFP$^+$, mCherry$^+$ and GFP$^+$mCherry$^+$ double positive cells, and performed qRT-PCR on sorted cell populations for *mgp*, a SC3 cluster 2 cell marker gene. Expression of *mgp* was selectively enriched in GFP$^+$mCherry$^+$ double positive cells (*Figure 4d*).

We next calculated differentially expressed genes between GFP$^+$mCherry$^+$ cells compared with the mCherry$^+$ cells using SCDE (*Kharchenko et al., 2014*). Based on the SCDE output genes were ranked and the ranked list was used with the WEB-based gene set analysis toolkit (WebGestalt) to explore the functional nature of the GFP$^+$mCherry$^+$ cells compared with the mCherry$^+$ cells

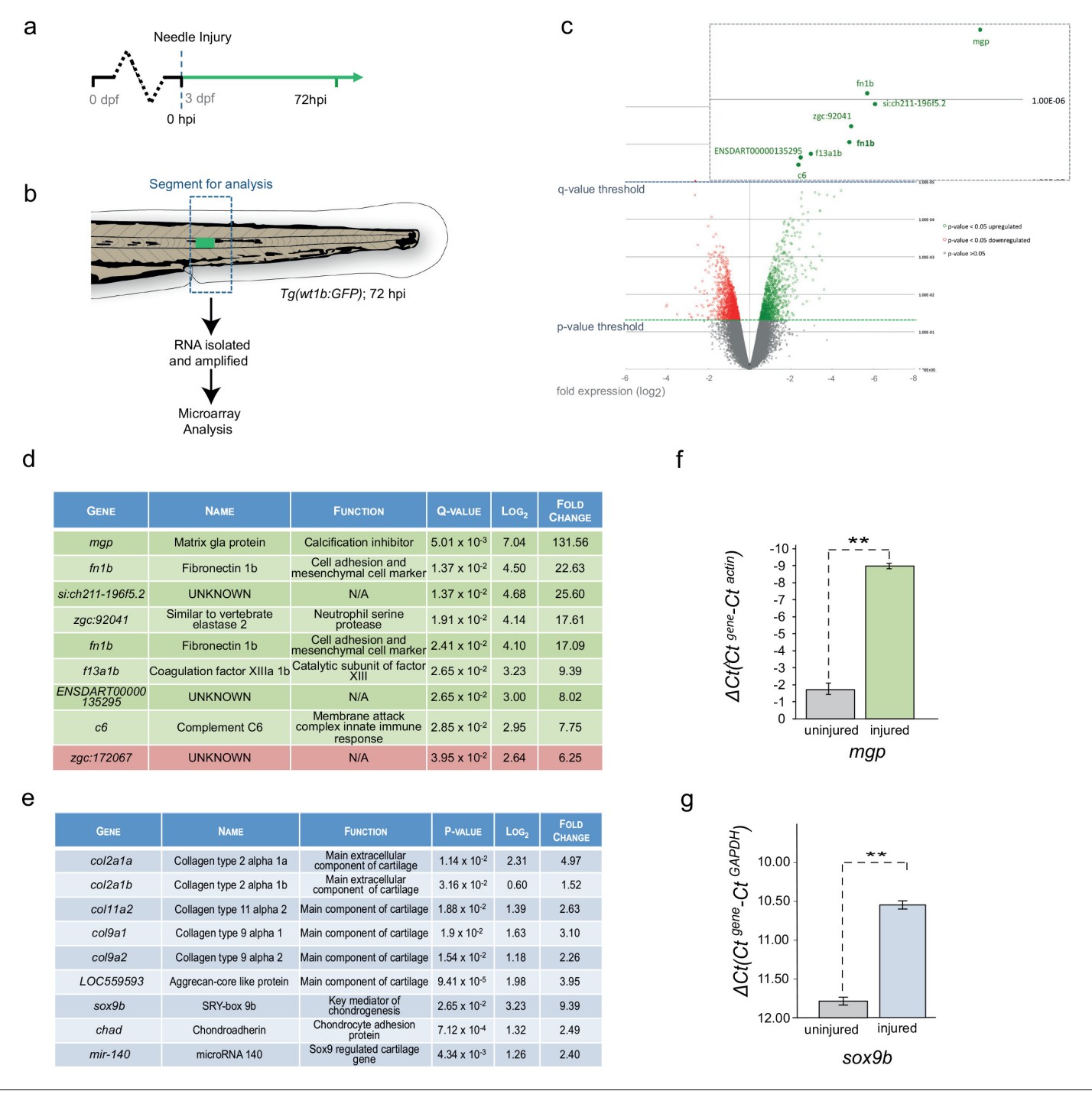

**Figure 3.** Cartilage genes are expressed in the notochord-injured zebrafish. (a) Experimental plan: 3 dpf *Tg(wt1b:gfp)* larvae were needle injured and grown for 72 hr with uninjured age-matched controls (n = 50 larvae per group). (b) The area around the *wt1b:gfp* expression was excised at 72 hpi (dotted area) and RNA was extracted and amplified. A similar area was taken from age-matched uninjured controls. (c) Volcano plot displaying the differentially expressed genes between injured and non-injured larvae. The y-axis measures the mean expression value of log 10 (p-value) and separates upregulated from downregulated genes. The x-axis represents the log2 fold change of expression. Significantly upregulated genes are shown as green circles or dots and downregulated genes are shown as red circles or dots. Green dotted line represents the p-value threshold (p<0.05) and blue dotted line represents the false discovery rate (FDR) or q-value threshold (q < 0.05). Genes with highest expression change are shown in magnified view. (d) Table showing the most significantly differentially expressed genes in injured larvae (q < 0.05). Upregulated genes are shown in green and downregulated genes are shown in red. (e) Table showing cartilage-associated genes that were significantly upregulated in the injured larvae (p<0.05). (f, g) Results of quantitative real-time PCR (qRT-PCR) of *mgp* and *sox9b*. The y-axis indicates the difference between the cycle threshold (Ct) value of

*Figure 3 continued on next page*

*Figure 3 continued*

the gene of interest and the Ct value of *β-actin* for *mgp* and *gapdh* for *sox9b*. Note that the y-axis is inverted to ease interpretation. Bars represent standard deviation from the mean. *mgp* **p=0.025; *sox9b* ***p=0.007; paired t-test; Experimental replicates: *mgp* = 2; *sox9b* = 1 at 48 hpi, and 1 at 72 hpi (40 embryos pooled per replicate). See Source Data files (*Figure 3—source data 1*; *Figure 3—source data 2*).
DOI: https://doi.org/10.7554/eLife.30657.010

The following source data is available for figure 3:

**Source data 1.** Raw data and statistical analyses for *Figure 3f*.
DOI: https://doi.org/10.7554/eLife.30657.011

**Source data 2.** Raw data and statistical analyses for *Figure 3g*.
DOI: https://doi.org/10.7554/eLife.30657.012

(*Figure 4e*). Expression of genes in signaling pathways, such as the TGF-ß pathway were reduced, while vacuolar and lysosomal pathway components were highly enriched in the GFP[+]mCherry[+] cells comparing gene sets from multiple databases. To explore the possibility of lysosome activity in more detail, we performed confocal imaging analysis of the wound site at 7 dpi and observed some GFP[+]mCherry[+] cells with large inclusions (presumably vacuoles), in the cytoplasm (*Figure 4f*). This suggests that some GFP[+]mCherry[+] cells may become vacuolated to replace those lost upon injury.

Next, given the expression of cartilage genes by microarray analysis, we performed gene set enrichment analysis (GSEA) with a list of zebrafish cartilage genes curated in AmiGO (*Supplementary file 1b, 1c*). Cartilage genes were significantly enriched in the cell cluster 2 (GFP[+]mCherry[+] cells) compared with cell cluster 3 (mCherry[+] cells), suggesting that it is specifically the *wt1b*-expressing sheath cells that express genes involved in cartilage formation (*Figure 4g*).

To explore the role of WT1 in the wound response, we compiled a list of WT1 target genes, and compared it with the rank order list of RNA transcripts expressed in the GFP[+]mCherry[+] cells by gene set enrichment analysis (GSEA) (*Supplementary file 1b, 1d*) (*Subramanian et al., 2005*). Unexpectedly, we discovered a set of WT1 regulated genes that were specifically repressed in the GFP[+]mCherry[+] cells (*Figure 4h*). WT1 can function with co-factors to repress or activate gene expression, and this new signature suggests that Wt1b may function as a repressor in the notochord damage response. Next, we performed gene expression analysis for all WT1 co-transcription factors described in (*Toska and Roberts, 2014*), and found *p53* to be most differentially expressed in GFP[+]mCherry[+] cells compared with mCherry[+] cells (*Figure 4i,j*). GSEA analysis showed that p53 target genes are enriched overall in the GFP[+]mCherry[+] cell populations (*Figure 4k*; *Supplementary file 1b, 1e*), however, when we specifically analysed the gene expression for those genes that were present in both the WT1 and p53 target gene list (*Supplementary file 1f*), we found a strong repression of genes that are regulated by both WT1 and p53 (*Figure 4l*). These data uncover an unexpected co-operation between Wt1b and p53 to negatively regulate a select subset of genes in the *wt1b*-expressing sheath cell subpopulation during the wound response.

## Vertebra form at the repair site via an unusual cartilage intermediate

The expression of cartilage genes in the wound tissue and in the *wt1b*-expressing sheath cell subpopulation suggests that the notochord wound may induce a previously unknown and alternative bone development process. We stained injured and control animals with alcian blue and alizarin red, which highlight cartilage and bone respectively. Cartilage was clearly visible at the site of injury as soon as three dpi (*Figure 5a*). This staining was significantly stronger and distinct from the highly coordinated segmental cartilage staining that normally occurs during larval development in the region of the future intervertebral discs, which is clearly visible in both injured and non-injured controls by 14 dpi (*Figure 5a*). Similarly, the alizarin red dye identified the anterior to posterior forming chordacentra rings during larval development. However, in injured zebrafish larvae, the normally uniform mineralization pattern was interrupted around the site of damage, leading to delayed formation of the chordacentra at later stages (*Figure 5a*). By 18 dpi, the injured site began to express bone matrix, and was visibly flanked by cartilage expressing segments (*Figure 5b*). This is unusual because during norm-physiological development of the vertebral elements, cartilage and bone stains mark distinct regions of the notochord.

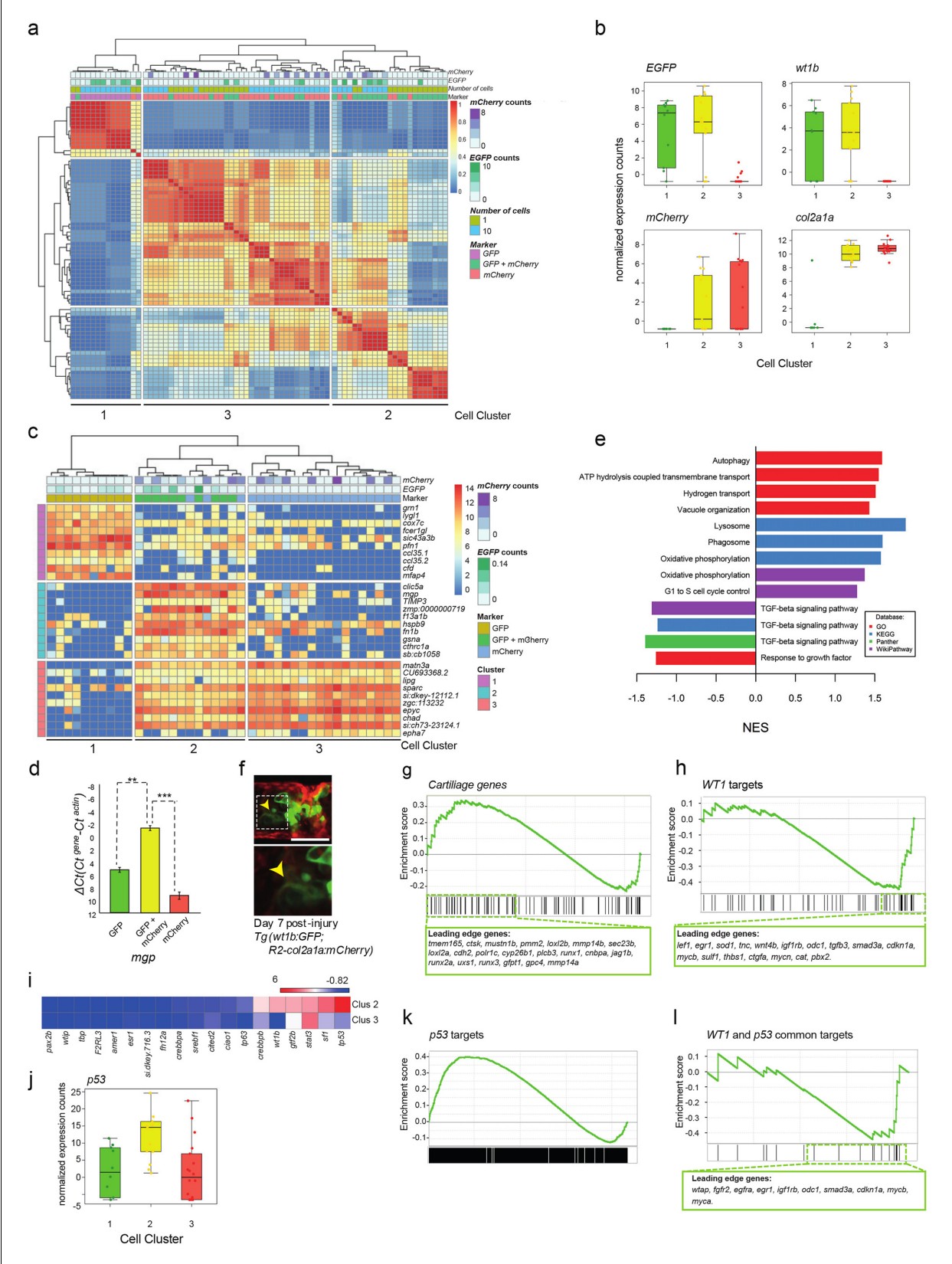

**Figure 4.** Single-cell and 10 cell sequencing of *wt1b*-shealth cell populations. (a) Single-cell and 10 cell SC3 unbiased clustering analysis reveals three distinct cell populations marked by *GFP* (cluster 1), *mCherry* (cluster 3), or *GFP* and *mCherry* (cluster 2). (b) *GFP*, *mCherry*, *wt1b* and *col2a1a* expression

*Figure 4 continued on next page*

*Figure 4 continued*

in 10 cell clusters. (c) Top 10 differential gene expression marker genes for 10 cell clusters. (d) Expression of *mgp* in different cell populations of injured zebrafish notochords. RNA was isolated from FACS sorted GFP, RFP and GFP/RFP expressing cells of the notochord of *Tg(wt1b:gfp; R2-cola2a1a: mCherry)* embryos, and gene expression was determined by qPCR. The y-axis indicates the difference between the cycle threshold (Ct) value of the gene of interest and the Ct value of beta-actin in injured and uninjured notochord. The y-axis is inverted for ease of interpretation. p-values are determined by paired t-test. Bars represent standard deviation. *mgp*: **p=0.035. Experimental replicates = 2. See Source Data file (*Figure 4—source data 1*). (e) Bar chart depicting functional analysis of differentially expressed genes between 10 cell SC3 cluster 2 and cluster three against five databases. Normalised enrichment score (NES, x-axis) calculated using online functional enrichment tool WebGestalt resource. Coloured bars match specific databases. (f) Images of the wound site seven days post injury in *Tg(wt1b:gfp;col2a1a:mCherry); nacre⁻/⁻* embryos. Arrows indicate vacuole-like structures. n = 7; experimental replicates: 1. Scale bar: 50 µm. (g) Gene set enrichment analysis (GSEA) of cartilage genes in *wt1*-expressing sheath cell (cluster 2) 10 cell group clusters (21 out of 82 genes were positively enriched; NES = 0.90). (h) GSEA of WT1 gene targets in *wt1b*-expressing sheath cell (cluster 2) 10 cell group clusters (19 out of 56 target genes were negatively enriched; NES = −1.44). (i) Heatmap of expression of WT1-interacting partners in 10 cell cluster 2 and cluster 3. (j) *p53* RNA expression in 10 cell clusters. (k) GSEA of p53 targets genes in *wt1b*-expressing sheath cell (cluster 2) 10 cell group clusters (358 out of 1442 genes were positively enriched; NES = 1.17). (l) GSEA of common p53 and WT1 gene targets in *wt1b*-expressing sheath cell (cluster 2) 10 cell group clusters (10 out of 19 genes were negatively enriched, NES = −1.11).

DOI: https://doi.org/10.7554/eLife.30657.013

The following source data and figure supplement are available for figure 4:

**Source data 1.** Raw data and statistical analyses for *Figure 4d*.
DOI: https://doi.org/10.7554/eLife.30657.015

**Figure supplement 1.** Quality control for the single-cell and 10 cell RNA sequencing.
DOI: https://doi.org/10.7554/eLife.30657.014

To evaluate the outcome of the injury in the ossification process, wild-type larvae were injured and stained using live calcein dye at 21 and 38 dpi (*Du et al., 2001*). The vertebrae that eventually formed were often smaller in a given space interval and appeared supernumerary compared with uninjured age-matched controls (*Figure 5c–e*).

The notochord patterns spine formation via the activation of various signals, and has been proposed to be an essential component of chordacentra formation (*Bensimon-Brito et al., 2012*; *Fleming et al., 2004*). Entpd5a (ectonucleoside triphosphate diphosphohydrolase 5) is an E-type NTPase that is expressed in osteoblasts and is essential for skeletal morphogenesis (*Huitema et al., 2012*). Recent evidence shows that metameric expression of *entpd5a* in notochord sheath cells is an essential requirement for the patterned formation of chordacentra rings (*LL-F and SS-M, personal communication*), with *entpd5a* expression serving as a readout for mineralizing activity (*Huitema et al., 2012*). We crossed the *Tg(wt1b:gfp)* transgenic line to a *Tg(entdp5a:pkRed)* line and followed the wound response. *wt1b* and *entpd5a* expressing cell populations were closely associated at the wound site indicating that mineralizing *entpd5a* cells may directly contribute to *wt1b⁺*-associated chordacentra response (*Figure 6a,b*).

Next, we wanted to explore the relationship between *entpd5a* expression domains and the vertebrae formation at the wound site. By 5 dpf, metameric *entpd5a* expression domains are clearly visible in the anterior notochord. We wounded the notochord in 5 dpf and 7dpf fish either in between two adjacent *entpd5a*-expression domains or aimed at the center of an *entpd5a*-expression domain. Fish that had been wounded between the *entpd5a*-expression domains appeared to have normal vertebrae structures at 25 dpi (n = 6/6). In contrast, damaging the *entpd5a*-expression domain led to a supernumerary vertebra at the wound site (n = 4/4; *Figure 6—figure supplement 1*).

Taken together, these results indicate that wounding alone is not sufficient to alter the vertebrae number, and that *entpd5a* expression domains likely play a role in vertebrae formation following injury. These experiments raise the possibility that the notochord wound assay at 3 dpf disrupts an as of yet unknown precursor cell population. Up-regulation of *entpd5a* at the damage site may be part of a patho-physiological wound repair response that disrupts and/or engages with a precursor cell population (such as the metameric *entpd5a* expression) leading to altered vertebra(e) in the adult.

## wt1b⁺ expression perdures into the adult vertebrae

We noticed that the *Tg(wt1b:gfp)* transgene expression was always associated with the site of vertebrae formation in the injured zebrafish that were raised to adulthood. To determine if *wt1b*

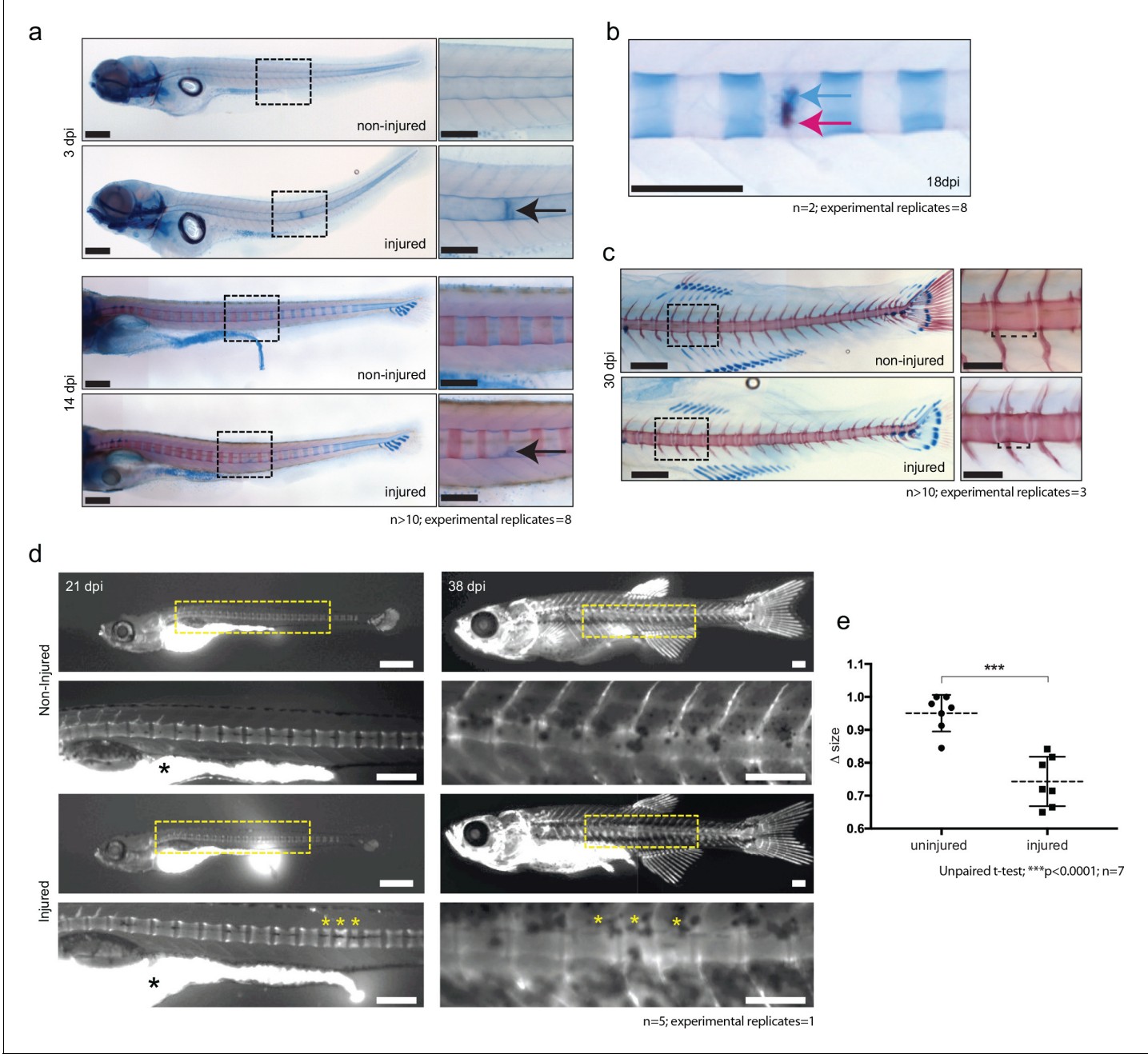

**Figure 5.** De novo bone formation occurs via a cartilage intermediate at the site of injury. (a) Alcian blue and Alizarin red staining at the site of injury in 3 and 14 dpi larvae. Ectopic cartilage deposit is indicated by arrow. n > 10; experimental replicates = 8. Scale bar left panels: 400 µm; scale bar right panels (zoomed images): 200 µm. (b) Alcian blue and Alizarin red staining at the site of injury at 18 dpi indicates the presence of bone and cartilage at the repair site (blue arrow = cartilage; red arrow = bone). n = 2; experimental replicates = 8. Scale bar: 200 µm. (c) Alcian blue and alizarin red staining of 30 dpi larvae reveals the formation of a smaller vertebra in the damaged area. n > 10; experimental replicates = 3. Scale bar left panels: 400 µm; scale bar right panels (zoomed images): 200 µm. (d) Live imaging of calcein stained zebrafish at 21 and 38 dpi in injured and uninjured fish. Vertebrae at damage site are indicated by yellow asterisks. Black asterisk denotes intestinal fluorescence. n = 5; experimental replicates = 1. Scale bar 21 hpf: 200 µm; scale bar 21 hpf zoomed: 100 µm; scale bar 38 hpf: 200 µm; scale bar 38 hpf zoomed: 100 µm. (e) The relative vertebra size difference (Δ size) between vertebrae at the site of injury (injured) and vertebrae in non-injured areas (uninjured). Vertebrae at the site of injury were significantly smaller than uninjured vertebrae (Unpaired t-test; ***p<0.0001 two-tailed; mean ±SEM uninjured larvae = 0.9506 + /- 0.02102 n = 7; mean ±SEM injured larvae = 0.7432 + /- 0.0284 n = 7; measurements taken at 30 and 38 dpi).

DOI: https://doi.org/10.7554/eLife.30657.016

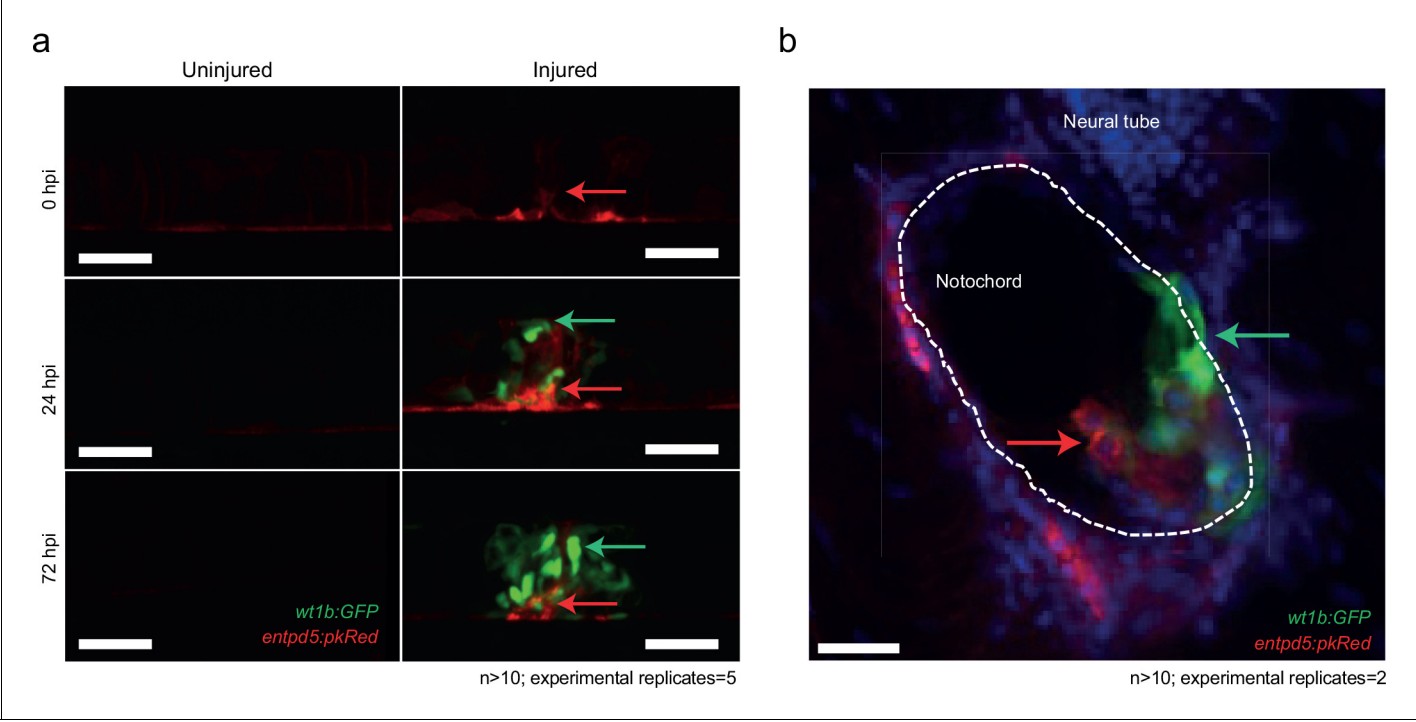

**Figure 6.** Distinct and closely associated *wt1b* and *entpd5a* subpopulations emerge at the damage site. (a) Live-imaging at the site of notochord injury in *Tg(wt1b:gfp; entpd5a:dkRed)* larvae. Expression of *wt1b:gfp* and *entpd5a:pkRed* at site of damage (green arrows and red arrows respectively) in injured and uninjured fish. n > 10; experimental replicates = 5. Scale bar: 50 μm. (b) Cryo-section of the injured area confirms distinct *wt1b:gfp* and *entpd5a:dkRed* subpopulations at site of damage. n > 10; experimental replicates = 2. Scale bar: 20 μm.

DOI: https://doi.org/10.7554/eLife.30657.017

The following figure supplement is available for figure 6:

**Figure supplement 1.** Needle damage of the *entpd5a* cell domain leads to supernumerary vertebrae.

DOI: https://doi.org/10.7554/eLife.30657.018

expression was transient at the wound, or sustained throughout the repair process, we raised needle injured *Tg(wt1b:gfp); casper* zebrafish larvae for up to 38 days.

GFP expression was sustained at the wound site, remaining in a small, cellular population at the site of damage, even as chordacentra developed and mineralized around the notochord over time (*Figure 7*). Small GFP expressing cells were further confirmed by α-GFP staining at the site of damage (*Figure 7b*). Strikingly, the *Tg(wt1b:gfp)* transgene maintained expression at this site up to 38 dpi (*Figure 7c,d,g*).

To gain a better understanding of how *wt1b:gfp* expressing cells engage with the newly forming vertebrae, we carried out live, confocal imaging of the area of damage (*Figure 7e–g*). The analysis revealed the presence of both fused and unfused vertebrae at the damaged site, and the sustained and strong expression of *wt1b:gfp* expressing cells associated with the developing vertebra at the repair site area (*Figure 7f*), even in fully formed spine structures (*Figure 7g*).

Taken together these results indicate that *wt1b:gfp* expressing cells both mark a subpopulation of cells that are rapidly activated at the site of the wound and also that these cells persist until adulthood, possibly orchestrating local vertebrae formation with wound repair.

## Discussion

We have uncovered wound-specific cellular heterogeneity in the zebrafish notochord that perdures throughout the wound healing process and during adult vertebra formation at the injury site (*Figure 8*). We discover that wounding leads to localized *wt1b* expression in the notochord sheath cells which then invade the site of the injury to form a stopper-like structure, likely to maintain notochord integrity. We show the specific de novo expression of *wt1b* in notochord sheath cells following

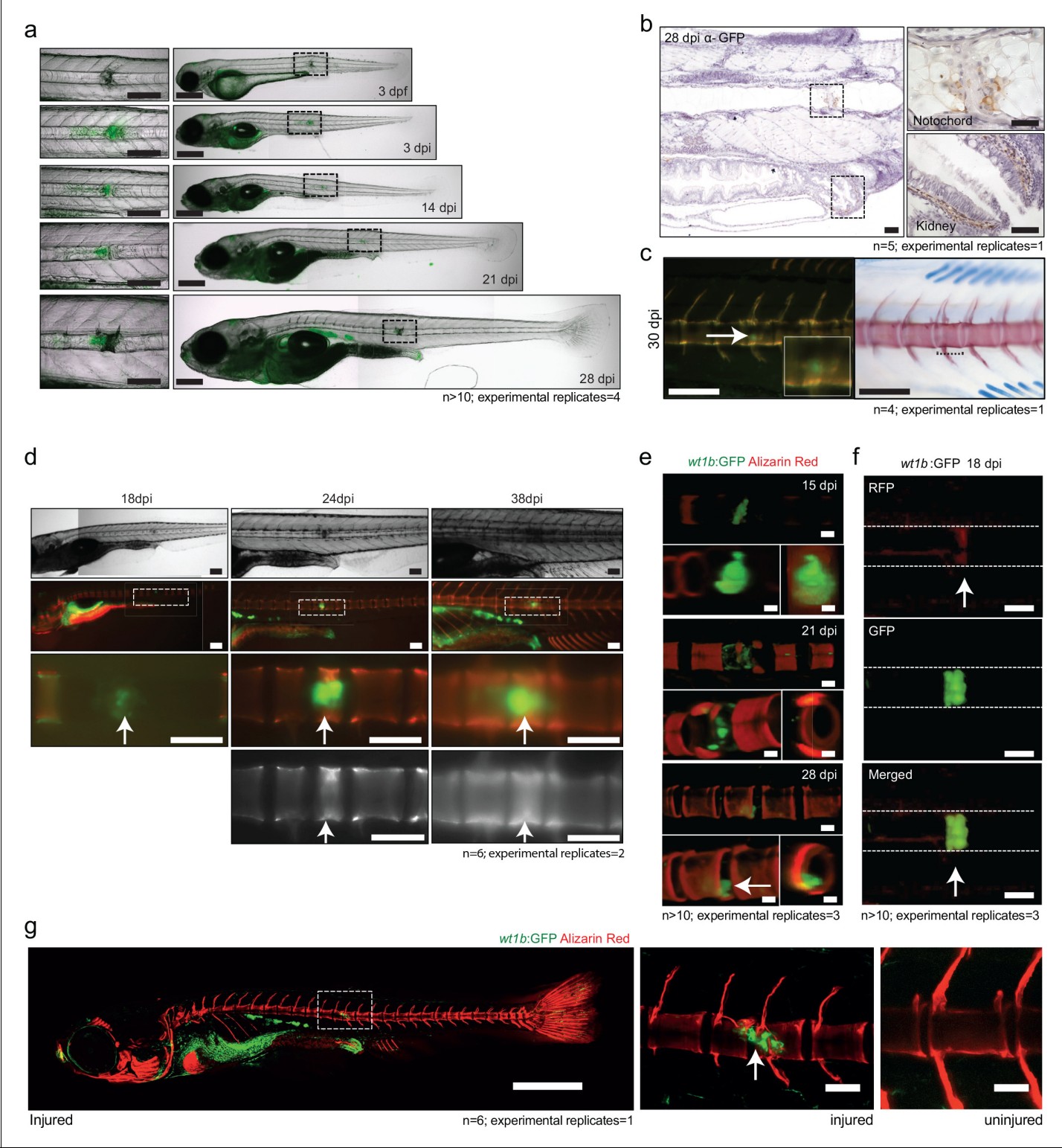

**Figure 7.** *wt1b* expressing cells are closely associated with vertebral development after injury. (a) Images of *Tg(wt1b:gfp)* zebrafish following needle injury at 3 dpf and raised to 28 dpi. n > 10; experimental replicates = 4. Scale bar left panels: 100 μm; scale bar right panels: 200 μm. (b) α-GFP staining of 28 dpi larvae at the site of the healing notochord wound and in the kidney. n = 5; experimental replicates = 1. Scale bar left panels: 50 μm. (c) Image of fish from *Figure 5a,c*, stained with alizarin red and imaged for *wt1b:gfp* expressing cells. GFP positive cells are found within the ectopic vertebra (white arrow and inset). n = 4; experimental replicates = 1. Scale bar left panels: 100 μm. (d) Long-term follow up of alizarin red stained *Tg(wt1b:gfp); casper* larvae shows that chordacentra formation is delayed around the site of injury. GFP cells mark the site of the future vertebra. n = 6; experimental

*Figure 7 continued on next page*

*Figure 7 continued*

replicates = 2. Scale bar: 100 µm; scale bar zoomed images: 50 µm. (**e**) Confocal imaging of 15, 21 and 28 dpi larvae reveals an overlapping expression between the *wt1b:gfp* expressing cells and the forming chordacentra (alizarin red stained) in the injured *Tg(wt1b:gfp); casper* larvae. n > 10; experimental replicates = 3. Scale bar: 100 µm. Imaging views are lateral, angled and cross-section view. (**f**) Confocal imaging highlights the overlapping presence of bone (alizarin red stained) and *wt1b:gfp* cells at the wound in 18 dpi larvae (arrow). n > 10; experimental replicates = 3. Scale bar: 100 µm. (**g**) Confocal scans of 24 dpi *Tg(wt1b:gfp)* larvae stained with alizarin red and expressing GFP at the injury site following notochord injury compared with uninjured control fish. GFP positive cells are present within the vertebrae at the injury site (arrow). Scale bar left fish: 1000 µm; scale bar on vertebrae images: 100 µm.

DOI: https://doi.org/10.7554/eLife.30657.019

wounding, despite an absence of *wt1b* expression during notochord development (*Figure 1e*,

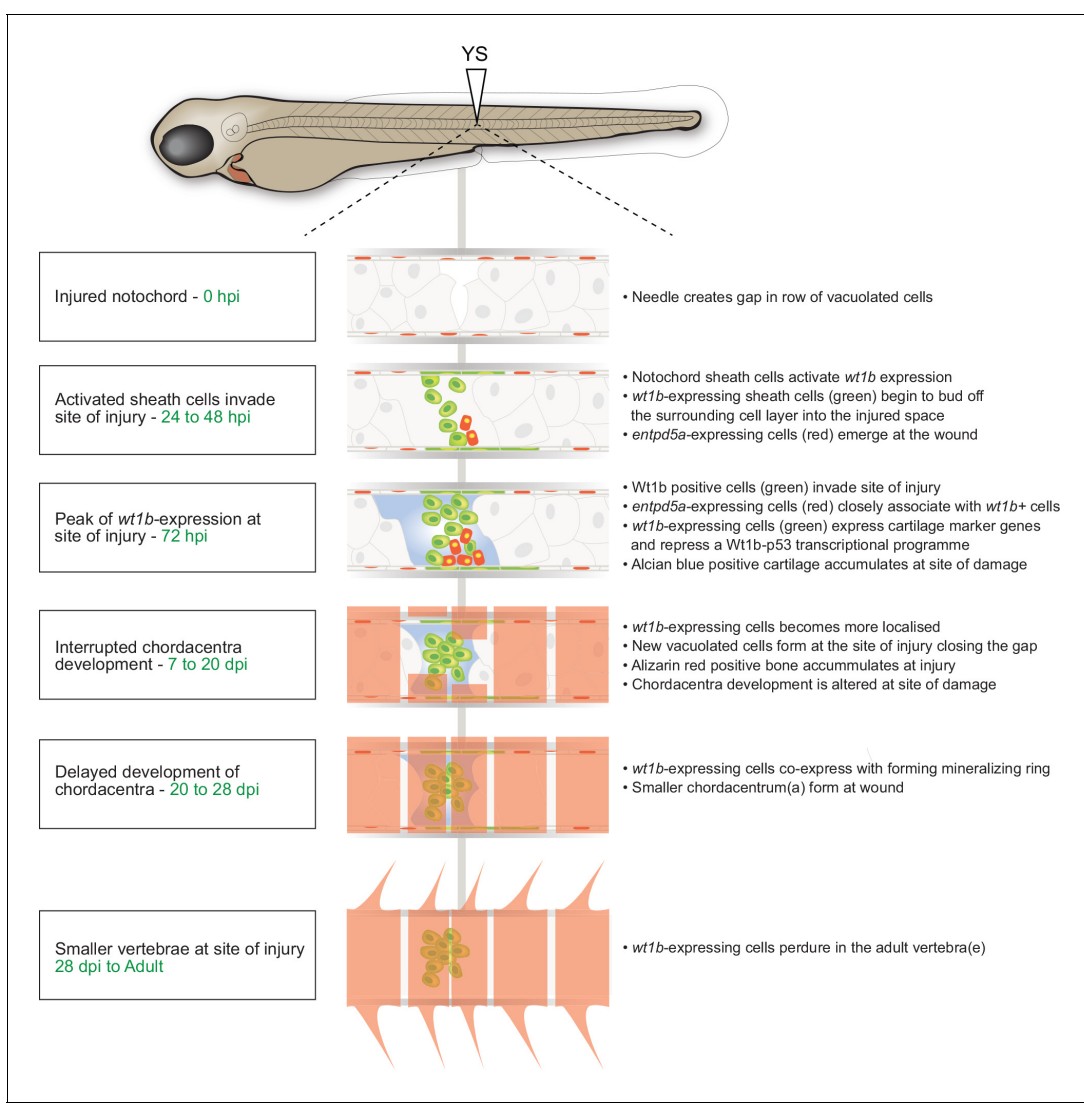

**Figure 8.** Schematic of the notochord wound response.

DOI: https://doi.org/10.7554/eLife.30657.020

The following source data and figure supplements are available for figure 8:

**Figure supplement 1.** Generation of *wt1b* mutant zebrafish.

DOI: https://doi.org/10.7554/eLife.30657.021

**Figure supplement 1—source data 1.** Raw data and statistical analyses for *Figure 8—figure supplement 1h*.

DOI: https://doi.org/10.7554/eLife.30657.022

*Figure 1—figure supplement 2*).

Very recently, Bagnat and colleagues reported the identification of notochord sheath cells involved in the replacement of vacuolated cells lost due to motion-dependent mechanical damage to the notochord in caveolin mutants (*Garcia et al., 2017*). In this context, sheath cells invade the vacuolated cell layer and differentiate into vacuolated cells to maintain turgor pressure. While we observe that most *wt1b*-expressing cells are tightly associated with a stopper-like (scar-like) structure from embryo to adult (*Figure 7*), we find some *wt1b* expressing cells appear vacuolated at the injury site at later stages (7 days post injury; *Figure 4f*), and that *wt1b*-expressing cells express vacuolar genes (*Figure 4e*). We also detected *entpd5a* expressing cell subpopulations at the wound that are distinct from *wt1b* expressing cells (*Figure 6*). These studies highlight a previously unknown complex and heterogeneous nature of the sheath cell populations, and suggest that the notochord sheath can sense and respond to different types of damage. Motion-dependent shear stress in *caveolin* mutants causes loss of vacuolated cells that are replaced by new vacuolated cells that arise from the sheath (*Garcia et al., 2017*), while acute damage (i.e. needle injury) that encompasses sheath and vacuolated cell damage, leads to sheath cells forming a seal that marks the site of new cartilage and vertebra (*Figure 8*). We show that *wt1b* expression marks a subpopulation of sheath cells in both damage responses (*Figure 1*, *Figure 2—figure supplement 2*), and suggest that additional factors are involved in the ultimate fate of *wt1b*-expressing cells (i.e. vacuolated cells versus scar like structure).

To address the function of Wt1b in the wound response, we generated a CRISPR-Cas9 genetic mutant that removes part of the C-terminal zinc-finger domains that are essential for WT1 function in mammalian systems. We find homozygous *wt1b* mutant zebrafish show no overt difference from wild type fish in the wound response (*Figure 8*; *Figure 8—figure supplement 1*). However, given the dramatic up-regulation of *wt1b* upon wounding, and given the continued expression up to adult stages, we consider it unlikely that Wt1b has no role in the process. Compensatory mechanisms have to be considered, and indeed, we find a small, but significant increase in *wt1a* in *wt1b*$^{\Delta5/\Delta5}$ wounded tissue. Furthermore, compensatory mechanisms downstream the Wt1b-p53 axis could mask a role, and further analysis beyond the scope of this study will be required to fully understand the functional significance of Wt1b in this subpopulation of cells.

By leveraging gene expression profiling, and single-cell and 10 cell sequencing of the wounded tissue, we discovered a mechanism for vertebra formation via a cartilage intermediate at the injury site. This is completely unexpected as in zebrafish, ossification of the chordacentra does not require the establishment of a cartige anlagen, but form via the direct mineralization of the fibrous notochord sheath (*Bensimon-Brito et al., 2012*; *Fleming et al., 2015*). The activation of *wt1b* in sheath cells that migrate towards the center of the notochord is reminiscent of the situation where *wt1b* expression is reactivated in epicardial cells that undergo EMT to produce vascular progenitors and migrate into the heart (*Martínez-Estrada et al., 2010*). This raises the question whether notochord sheath cells may also be mesothelial in nature and if the invading *wt1b* expressing cells are produced via an EMT or, perhaps more accurately, a mesothelial to mesenchyme transition. While *wt1b*-positive cells express some mesenchymal genes (*Figure 3d*), we did not find evidence that these cells express classical gene signatures related to known EMT processes in the damaged tissue. This may be evidence of an as of yet unknown process in the wound response, or possibly because the EMT process was primarily completed by the time of our analysis at 3 days post injury.

Surprisingly, we have uncovered a new Wt1b-p53 gene expression signature that is specifically repressed in *wt1b*$^+$ sheath cells (*Figure 4k*). p53 is a transcription factor that in addition to its well-established role as a tumor suppressor, functions to inhibit premature osteoblast differentiation and bone remodeling (*Liu and Li, 2010*). Several lines of evidence support a direct WT1-p53 interaction, and that p53 can modify activity of WT1 transcriptional activity from an activator to a repressor on select promoters in vitro (*Maheswaran et al., 1995*; *Maheswaran et al., 1993*). However, the in vivo function for the WT1-p53 interaction is not yet understood, and loss of *p53* in *wt1*-null mutant mice does not alter the *wt1*-null phenotype (*Menke et al., 2002*). Here, we identify a Wt1b-p53 axis specifically in the repair of a notochord wound. The Wt1b-p53 gene signature includes repression of genes that regulate osteogenesis in mammals, including *myc* (*a* and *b*), *egr1* and *igfrb* (*Piek et al., 2010*; *Reumann et al., 2011*; *Wang et al., 2015*). We propose that repair-specific transcription factors participate in notochord healing by co-ordinating expression of cartilage genes such as *sox9* and *mgp* (*Schurgers et al., 2013*; *Sweatt et al., 2003*; *Zebboudj et al., 2002*), with a Wt1b-p53

transcriptional axis repressing premature expression of osteogenesis genes in the first few days following wounding. We see *entpd5*[+] notochord sheath cells in the wound area (*Figure 6*), and since *entpd5* is essential for mineralization, it seems likely that these cells, in conjunction with cartilage formation at the site of injury, play a role in centrum formation (*Figure 8*). Eventually, smaller vertebra form at the wound site, and *wt1b:gfp* cells remain tightly associated with this/these vertebra(e) into adulthood. This mode of notochord wound healing and vertebra formation may be a salvage structure to effectively maintain structural integrity of the developing axial skeleton.

# Materials and methods

## Key resources table

| Reagent type or resource | Designation | Source or reference | Identifiers | Additional information |
|---|---|---|---|---|
| Gene (Danio Rerio) | *sagff214a* | NA | ZFIN ID: ZDB-ALT-110315–2 | |
| Gene (Danio Rerio) | *wt1a* | NA | ZFIN ID: ZDB-GENE-980526–558 | |
| Gene (Danio Rerio) | *col2a1a* | NA | ZFIN ID: ZDB-GENE-980526–192 | |
| Gene (Danio Rerio) | *entpd5a* | NA | ZFIN ID: ZDB-GENE-100419–1 | |
| Gene (Danio Rerio) | *sox9b* | NA | ZFIN ID: ZDB-GENE-001103–2 | |
| Gene (Danio Rerio) | *wt1b* | NA | ZFIN ID: ZDB-GENE-050420–319 | |
| Genetic reagent (Danio Rerio) | Tg(entpd5:kaede) | *Geurtzen et al., 2014* doi: 10.1242/dev.105817 | ZFIN ID: ZDB-ALT-150223–1: hu6867 | Same BAC used as *Huitema et al. (2012)* (DOI: 10.1073/pnas.1214231110) with kaede insertion at first translated ATG |
| Genetic reagent (Danio Rerio) | Tg(entpd5:pkRed) | This paper | ZFIN ID: hu7478 | Same BAC used as *Huitema et al. (2012)* (DOI: 10.1073/pnas.1214231110) with pkRed insertion at first translated ATG |
| Genetic reagent (Danio Rerio) | Tg(SAGFF214a;UAS:gfp) | *Yamamoto et al. (2010)* DOI: 10.1242/dev.051011 | ZFIN ID: ZDB-FISH-150901–18089 | |
| Genetic reagent (Danio Rerio) | Tg(wt1b:GFP,R2col2a1a:mCherry) | This paper | ZFIN ID: ZDB-ALT-180105–1; zfin.org:ue401Tg | |
| Genetic reagent (Danio Rerio) | Tg(wt1a:GFP) | *Bollig et al. (2009)* DOI: 10.1242/dev.031773 | ZFIN ID: ZDB-FISH-150901–2540 | |
| Genetic reagent (Danio Rerio) | Tg(wt1b:GFP) | *Perner et al. (2007)* DOI: 10.1016/j.ydbio.2007.06.022 | ZFIN ID: ZDB-FISH-150901–1774 | |
| Genetic reagent (Danio Rerio) | casper | *White et al. (2008)* DOI: 10.1016/j.stem.2007.11.002 | ZFIN ID: ZDB-ALT-990423–22 | |
| Genetic reagent (Danio Rerio) | zebrafish codon optimised cas9 mRNA | *Jao et al. (2013)* DOI: 10.1073/pnas.1308335110 | | |
| Genetic reagent (Danio Rerio) | Wt1b p.F319fsX321 | this paper | ZFIN ID: ZDB-ALT-180105–2; zfin.org:ue402 | zebrafish wt1b mutant line, mutation is in the exon coding the zinc finger 2 |
| Genetic reagent | Tol2 transposase | *Kawakami, 2007* DOI: 10.1186/gb-2007–8 s1-s7 | | |

*Continued on next page*

*Continued*

| Reagent type or resource | Designation | Source or reference | Identifiers | Additional information |
|---|---|---|---|---|
| Antibody | anti-WT1 (rabbit polyclonal) | This paper, Cambridge Research Biochemicals antibody production services | | (1:25000); anti-WT1 was designed using the TARGET antibody production protocol from Cambridge Research Biochemicals using a conserved protein sequence from the C-terminal of the zebrafish Wt1a and Wt1b proteins. |
| Antibody | AlexaFluor 488 antibody (rabbit polyclonal) | Invitrogen | Donkey anti-Rabbit IgG (H + L) Secondary Antibody, Alexa Fluor 488: R37602; RRID:AB_221544 | (1:800) |
| Antibody | anti-GFP (rabbit polyclonal) | Cell Signaling Technology | Cell Signaling Technology: GFP Antibody (Rabbit): 2555S; RRID:AB_10692764 | (1:1500) |
| Recombinant DNA reagent (plasmid) | R2-col2a1a:mCherry | *Dale and Topczewski (2011)* DOI: 10.1016/j.ydbio.2011.06.020 | | |
| Sequence-based reagent | wt1b mutant sgRNA | this paper | | GGTCAGACCTGGAGAAGCGG |
| Commercial assay or kit | Dako REAL EnVision Detection System kit | Dako | Dako REAL EnVision Detection System, Peroxidase /DAB+, Rabbit/Mouse: Code K5007 | |
| Commercial assay or kit | Low Input Quick Amp Labelling Kit | Agilent Technologies | Low Input Quick Amp Labeling Kit, one-color: 5190–2305 | |
| Commercial assay or kit | Nextera XT DNA Library Preparation Kit (96 samples), | Illumina | Nextera XT DNA Library Preparation Kit (96 samples),: Cat: FC-131–1096 | |
| Commercial assay or kit | 4 × 44K Whole Zebrafish (V3) Genome Oligo Microarray | Agilent Technologies | | |
| Chemical compound, drug | DPX Mountant for histology | Sigma-Aldrich | DPX Mountant for histology: 06522–100 ML | |
| Chemical compound, drug | ProLong Gold Antifade Mountant with DAPI | Invitrogen | ProLong Gold Antifade Mountant with DAPI: P36931 | |
| Chemical compound, drug | Trizol | Invitrogen | TRIzol Reagent: 15596026 | |
| Chemical compound, drug | FACSmax cell disassociation solution | Genlantis | FACSmax Cell Dissociation Solution: AMS.T200110 | |
| Chemical compound, drug | OCT compound Tissue-Tek | Sifam Instruments LTD | OCT COMPOUND TISSUE-TEK: SIFAAGR1180 | |
| Chemical compound, drug | Nystatin | Sigma-Aldrich | Nystatin powder, BioReagent, suitable for cell culture: N6261-500KU | |
| Software, algorithm | Color Inspector 3D | ImageJ 1.51 n plugin | RRID:SCR_002285 | |
| Software, algorithm | Fiji | ImageJ 1.51 n | RRID:SCR_002285 | |
| Software, algorithm | Feature Extraction Software | Agilent Technologies | RRID:SCR_014963 | |
| Software, algorithm | Rsubread package | R-3.3.3; *Liao et al. (2013)*. DOI: 10.1093/nar/gkt214 | RRID:SCR_009803 | |
| Software, algorithm | SCDE | *Kharchenko et al. (2014)* DOI: 10.1038/nmeth.2967 | RRID:SCR_015952 | |

*Continued on next page*

*Continued*

| Reagent type or resource | Designation | Source or reference | Identifiers | Additional information |
|---|---|---|---|---|
| Software, algorithm | SC3 package | *Kiselev et al. (2017)* DOI: 0.1038/nmeth.4236 | RRID:SCR_015953 | |
| Software, algorithm | Scater package | *McCarthy et al., 2017* DOI: 10.1093/bioinformatics/btw777 | RRID:SCR_015954 | |
| Software, algorithm | STAR RNA-seq aligner | *Dobin et al. (2013)* DOI: 10.1093/bioinformatics/bts635 | RRID:SCR_015899 | |
| Software, algorithm | FACSDiva software | Version 6.1.3; BD Biosciences | RRID:SCR_001456 | |
| Software, algorithm | Webgestalt | *Wang et al. (2013)* DOI: 10.1093/nar/gkt439 | RRID:SCR_006786 | |
| Software, algorithm | Rosetta Resolver gene expression data analysis system | Rosetta Biosoftware | RRID:SCR_008587 | |
| Other | Alizarin Red | Fisher Scientific | Alizarin Red S Sodium Salt25G:11329707 | |
| Other | Alcian Blue | Sigma | Alcian Blue 8Gx: A5268-10G | |

## Zebrafish lines

All experimental procedures were approved by the University of Edinburgh Ethics Committee and were in accordance with the UK Animals (Scientific Procedures) Act 1986. Transgenic lines for this study include: *Tg(entpd5a:pkRed)* (*Huitema et al., 2012*), *Tg(SAGFF214A:GalFF;UAS:gfp)* (*Yamamoto et al., 2010*), *Tg(wt1a:gfp)* (*Bollig et al., 2009*), *Tg(wt1b:gfp)* (*Bollig et al., 2009*; *Perner et al., 2007*). Many of the studies were performed in a transparent background created by crossing homozygous *Tg(wt1b:gfp)* fish to homozygous pigment-free transparent *casper* fish (*White et al., 2008*). The *Tg(wt1b:gfp;R2col2a1a:mCherry)* line was created by injecting the *R2col2a1a:mCherry* construct (*Dale and Topczewski, 2011*) with a Tol2 transposase (*Kawakami, 2007*) into *Tg(wt1b:gfp;casper)* zebrafish embryos, generating *Tg(R2col2a1a:mCherry)*[ue401Tg].

## Notochord needle injury and tail amputation assays

For notochord wounds on day 3, larvae were anaesthetised in tricaine, placed sagittally on a petri dish and either inserted gently with an electrolysis-sharpened tungsten wire or tail amputated at different levels. Injured larvae were transferred to fresh water to recover and observe. Non-injured age-matched larvae were grown as non-injured controls. For injuries on day 5 and 7 pf larvae, the notochord wounds were generated using stainless steel insect pins (0.10 mm), under fluorescence light in a Leica (Germany) M165FC with a 1.0X plan Apo objective. All pictures (brightfield, Kaede and alizarin red stains) where taken using an Olympus (Japan) szx16 with a 1.5X Plan Apo objective with a Leica DFC 450C camera.

## Whole-mount microscopy

Live and fixed whole-mount time-course and time-lapse experiments were performed using an AZ100 upright macroscope (Nikon; Japan) using a x2 and x5 lens with a Retiga Exi camera (Qimaging) or Coolsnap HQ2 camera (Photometrics; Tucson, Arizona) or a Leica MZFLIII fluorescence stereo microscope fitted with a Qimaging Retiga Exi camera. Images were analyzed and processed using the IPLab Spectrum and Micro-Manager software. Live and fixed whole-mount confocal imaging was performed using an A1R confocal system (Nikon) using x10 and a x20 lens over a Z-plane range of 80–100 µm (approximate width of the notochord) using a 480 nm laser (GFP), a 520 nm (RFP) and/or a 561 nm laser (alizarin red). Images were captured and analysed using Nis-Elements C software (Nikon). Images of the nystatin-treated larvae were acquired by using a 20x lens on the Imaging Platform Dragonfly (Andor Technologies, Belfast UK) with 488 nm (GFP) and 561 nm (RFP) lasers built on a Nikon TiE microscope body with a Perfect focus system (Nikon Instruments). Z stacks through the notochord were collected in Spinning Disk 25 µm pinhole mode on the Zyla 4.2 camera using a Bin of 1 and frame averaging of 1 using Fusion v1.4 software. Data were visualised using Fiji, and

histograms generated using its Color Inspector 3D plugin. Multiphoton confocal time-lapse imaging was performed using an SP5 confocal microscope (Leica) equipped with a Ti:Sapphire multiphoton laser (Spectra Physics; Santa Clara, California) and a three axis motorised stage. For confocal imaging and time-lapse experiments, anaesthetised injured and non-injured larvae were embedded sagittally in a drop of 1% low-melting point agarose prior to imaging, in a specially designed glass insert, which was covered in a mixture of E3 medium and anaesthetic. All time-lapse imaging was done at 30 or 60- min intervals over 48 hr using an incubation chamber (Solent Scientific; UK) under a constant temperature of 28°C and larvae were terminated in an overdose of tricaine at the end of each experiment.

## Histology

Zebrafish larvae younger than 20 dpf were culled and fixed overnight in 4% PFA/PBS at 4°C. The fixed larvae were washed in PBS, dehydrated in rising methanol/PBS concentrations and cleared in xylene before being paraffin embedded for sectioning. Older zebrafish were culled and fixed in 4% PFA/PBS at 4°C for 3 days with an abdominal incision to ensure tissue penetrance of the fixative (*Walker and Kimmel, 2007*; *Wojciechowska et al., 2016*). Fish were decalcified using 0.5M EDTA (pH 7.5) for 5 days in a rocker at 4°C and dehydrated in 70% ethanol at 4°C. Fish were embedded in paraffin using a Miles Scientific Tissue TEK VIP automated processor. Embedded larvae and older zebrafish were sectioned using a Leica RM2235 rotary microtome to a width of 5 µm. Sections were haematoxylin and eosin (H and E) stained and mounted using DPX mountant for histology (Sigma-Aldrich; St. Louis, Missouri). For cryosections, zebrafish larvae were embedded in OCT compound Tissue Tek (Sifam Instruments LTD; UK) and cut to 8 µm following protocols available at www.zfin.org.

## Wt1 zebrafish antibody

The Wt1 antibody was synthesised by Cambridge Research Biochemicals (CRB; UK) antibody production services (http://www.crbdiscovery.com/home). The antibody was created using the CRB TARGET antibody production protocol (https://www.crbdiscovery.com/antibodies/target-antibodies/), which used a HPLC-purified peptide made from the third zinc finger domain of zebrafish Wt1 (CQRKFSRSDHLKTHTRT) to immunise two rabbits. This epitope is found in both Wt1a and Wt1b, and the antibody is expected to detect both zebrafish Wt1a and Wt1b. The serum from each rabbit was collected at multiple time points and tested for the presence of Wt1 antibodies using an electrophoretic mobility shift assay (EMSA). The purified polyclonal antibody was extracted from the rabbit serum on the final collection day. Western blot analysis of lysates from zebrafish (24 hpf) revealed a strong band at approximately 45 kDa, consistent with the size of zebrafish Wt1a/b protein. Immunofluorescence on paraffin-embedded sections with Wt1 antibody (diluted 1:33,000) revealed cell-specific staining in the kidney and notochord wound site that was depleted by co-incubation of the Wt1 antibody with the Wt1 epitope peptide.

## Immunohistochemistry

Slides were de-waxed in xylene and rehydrated through decreasing ethanol washes, before being incubated in a bleach solution to remove pigment. Antigen-unmasking was performed as previously described (*Patton et al., 2005*). with the Dako REAL EnVision Detection System kit (Dako; UK) following manufacturer's instructions. Slides were incubated overnight at 4°C with the following antibodies: anti-rabbit α-GFP (1:1,500; Cell Signaling Technology) and anti-rabbit α-WT1 (1:25,000; Cambridge Research Biochemicals; UK). An Axioplan II fluorescence microscope (Zeiss; Germany) with a Plan Apochromat objective was used for brightfield imaging of tissue sections. Images were captured using a Qimaging Micropublisher 3.3mp cooled CCD camera and analysed using the IPLab Spectrum software.

## Immunofluorescence

Slides were processed as described above and blocked in 10% heat inactivated donkey serum for 2 hr. Slides were incubated overnight at 4°C with α-WT1 (1:33,000) antibody diluted in 1% heat inactivated donkey serum in TBSTw. Slides were incubated for 1 hr in a secondary anti-rabbit AlexaFluor 488 antibody (1:800) (Invitrogen; Carlsbad, California) in 1% heat inactivated donkey serum and

mounted in ProLong Gold Antifade Mountant with DAPI (Invitrogen) overnight before being imaged in a fluorescent stereomicroscope.

## Tissue staining

Live bone staining was performed using 0.2% (w/v) calcein or using 50 µg/ml alizarin red (Fisher Scientific; UK) as previously described (*Kimmel et al., 2010*). For cartilage and bone staining, we used alcian blue and alizarin red following the protocol outlined in (*Walker and Kimmel, 2007*) with modifications from protocols on www.zfin.org. Bone and cartilage staining in fixed larvae was performed on PFA fixed and then methanol dehydrated specimens, treated overnight at 4°C with 0.02% (weight to volume) alizarin red in 70% ethanol. Specimens were bleached ($H_2O_2$) and cleared before storing in glycerol for imaging.

## RNA extraction and microarray analysis

*Tg(wt1b:gfp)* zebrafish larvae were needle injured and grown to 72 hpi with age-matched non-injured controls. The area around the site of injury was dissected and transferred into 1 ml of chilled RNA-later. The samples were centrifuged into a pellet at 4°C and macerated in 500 µl of Trizol (Invitrogen) using a 25G $^{5/8}$1 ml syringe. RNA was extracted following Trizol manufacturer's instructions and eluted into 15 µl of distilled $H_2O$. Extracted RNA was sent to Myltenyi Biotec (Germany) who conducted the microarray analysis. Injured and non-injured samples were sent in triplicates and the RNA was amplified and Cy3-labelled using a Low Input Quick Amp Labelling Kit (Agilent Technologies; UK) following manufacturer's instructions. The labelled cRNA was hybridised against a 4 × 44K Whole Zebrafish (V3) Genome Oligo Microarray (Agilent Technologies). The microarray images were processed using the Feature Extraction Software (FES – Agilent Technologies) and differential gene expression was determined using the Rosetta Resolver gene expression data analysis system (Rosetta Biosoftware).

## Fluorescence-Activated cell sorting

The trunk region of fifty *Tg(wt1b:gfp; R2col2a1a:mCherry)* injured larvae and non-injured 72 hpi larvae were dissected and collected separately in cold PBS + 2% fetal calf serum (FCS). Tissue disassociation was adapted from a previously described protocol (*Manoli and Driever, 2012*) and centrifuged cells were collected in FACSmax cell disassociation solution (Genlantis; San Diego, California). The samples were passed twice through a 40 µm cell strainer, collected in an agar-coated petri dish on ice and transferred into an eppendorf tube to be sorted by a FACSAria2 SORP instrument (BD Biosciences; UK) equipped with a 405 nm, a 488 nm and a 561 nm laser. Green fluorescence was detected using GFP filters 525/50 BP and 488 nm laser, red fluorescence was detected using 585/15 BP filter and 561 nm laser. Data were analysed using FACSDiva software (BD Biosciences) Version 6.1.3. For single cell sequencing, single cell or 10 cells were sorted into 96-well plates; for quantitive realtime PCR (qPCR) analysis, cells were collected by centrifuging at 6000 rpm for 5 min.

## *wt1b* mutant line

The *wt1b* genetic mutant line was generated by CRISPR/Cas9 with a guide RNA target GGTCA-GACCTGGAGAAGCGG (on the reverse strand) in the exon of *wt1b* that encodes zinc finger 2. Crispr/Cas9 genome editing was carried out following the Joung lab protocol (*Hwang et al., 2013*), with injection of a zebrafish codon optimized Cas9 mRNA (*Jao et al., 2013*). Two founders carrying germline mutations at the target site were identified: one mutation is a deletion of 12 bp (leading to an in-frame deletion) and the other is a deletion of 5 bp (*wt1b*$^{\Delta 5}$)$^{ue402}$, and was used in this study.

## qRT-PCR analysis

RNA extraction, in vitro synthesis and PCR amplification of cDNA were performed using the Smart Seq2 protocol (*Picelli et al., 2014*). Amplified cDNA was quantified using a bioanalyzer, and directly used for qRT-PCR without further dilution of the cDNA template. qPCR was performed in a Roche LightCycler480 using a SYBR green protocol. ΔCt (the difference between the cycle threshold (Ct) value of the gene of interest and the Ct value of *ß-actin* or *gapdh*) was used to compare the

expression level of genes. Statistics (St Dev and paired T-test) were performed using Matlab (Natick, Massachussets). Primers are listed in *Supplementary file 2*.

## Single-cell and 10 cell sequencing

RNA extraction, in vitro synthesis and PCR amplification of cDNA, and construction of a sequencing library using the Nextera XT DNA Library Preparation Kit (Illumina; San Diego California) according to the Smart Seq2 protocol with minor modifications as described before (*Picelli et al., 2014*). Libraries were sequenced on a NextSeq Illumina sequencer. Reads were mapped against the Ensembl *Danio rerio* reference genome version GRCz10.90 (*Ensembl, 2017*) with the inclusion of the reference for the spike in controls from the ERCC consortium, as well as the coding sequence for EGFP and mCherry, using STAR RNA-seq aligner (*Dobin et al., 2013*). For quality control and pre-processing, quantification of mapped reads per gene was calculated using the Rsubread package in R-3.3.3 (*Liao et al., 2013*). Genes that were not expressed in any cells were excluded. The gene counts were loaded as a scater object in R-3.3.3 (using the scater package) and standard quality control metrics were calculated (*McCarthy et al., 2017*). Quality control exclusion criteria were cells with more than 25% of reads mapping to ERCCs or fewer than 100,000 reads or fewer than 1000 genes detected (at least one read per gene) were rejected (see *Figure 4—figure supplement 1* and *Supplementary file 1a*).

Consensus clustering set to three clusters was conducted on the single and 10 cells using the SC3 package (*Kiselev et al., 2017*). The 10 cell group was isolated and SC3 consensus clustering set to three clusters was conducted on these cells alone. Differential expression between cluster 2 and cluster 3 of the SC3 10 cell analysis was conducted using SCDE (*Kharchenko et al., 2014*). A differential expression list, ranked from cluster 2 to cluster three according to z-score was used for the GSEA analysis (*Mootha et al., 2003*; *Subramanian et al., 2005*). The differential expression list was tested against gene lists compiled from online resources (*Supplementary file 1b*). Functional analysis between the ranked 10 cell list and online gene lists for gene ontology (biological processes, non-redundant) and pathways (KEGG, Panther, Reactome and WikiPathway databases were used) using the online tool WebGestalt and gene set enrichment function (*Wang et al., 2013*).

## Vertebrae size measurements and statistical analysis

The vertebrae size difference in injured zebrafish larvae (age range 30 dpi to 38 dpi) were compared between vertebrae at the site of injury (injured) and vertebrae outside of the site of injury (uninjured). Injured vertebrae and uninjured vertebrae were measured and the average length was recorded for each group. The average lengths were then compared and the relative size difference was calculated. The relative size difference between each group (injured:uninjured vs. uninjured:uninjured) was compared using an unpaired t-test.

## Acknowledgements

We thank staff at the MRC Human Genetics Unit for excellent support, including Craig Nicol and Connor Warnock for assistance with figure design, Elisabeth Freyer for FACS analysis, and the zebrafish facility staff in Edinburgh and Utrecht for zebrafish husbandry. We thank Lee Murphy at the Edinburgh Clinical Research Facility for excellent service and support, Jeanette Baran-Gale for important advice on the bioinformatics analysis, and Andrea Coates for critical reading of the manuscript.

## Additional information

### Funding

| Funder | Grant reference number | Author |
| --- | --- | --- |
| Medical Research Council | MC_PC_U127585840 | Juan Carlos Lopez-Baez<br>Zhiqiang Zeng<br>Alessandro Brombin<br>Witold Rybski<br>E Elizabeth Patton |

| Medical Research Council | MC_PC_U127527180 | Juan Carlos Lopez-Baez<br>Nicholas D Hastie |
|---|---|---|
| Medical Research Council | Doctoral Training Programme in Percision Medicine | Daniel J Simpson |
| H2020 European Research Council | ZF-MEL-CHEMBIO - 648489 | Hannah Brunsdon<br>Alessandro Brombin<br>E Elizabeth Patton |
| Medical Research Council | Discovery Award MC_PC_15075 | Angela Salzano |
| Melanoma Research Alliance | 401181 | Alessandro Brombin<br>E Elizabeth Patton |
| L'Oreal USA | 401181 | Alessandro Brombin<br>E Elizabeth Patton |
| Japan Society for the Promotion of Science | 15H02370 | Koichi Kawakami |
| Japan Agency for Medical Research and Development | National BioResource Project | Koichi Kawakami |
| Leibniz-Gemeinschaft | | Christoph Englert |
| University Of Edinburgh | Chancellor's Fellowship | Tamir Chandra |
| Cells in Motion - Cluster of Excellence | EXC 1003-CiM | Stefan Schulte-Merker |

The funders had no role in study design, data collection and interpretation, or the decision to submit the work for publication.

## Author contributions

Juan Carlos Lopez-Baez, Conceptualization, Resources, Data curation, Formal analysis, Validation, Investigation, Visualization, Methodology, Writing—original draft, Writing—review and editing; Daniel J Simpson, Conceptualization, Data curation, Software, Formal analysis; Laura LLeras Forero, Conceptualization, Formal analysis, Validation, Investigation; Zhiqiang Zeng, Resources, Generating transgenic lines and molecular biology tools; Hannah Brunsdon, Conceptualization, Formal analysis, Investigation; Angela Salzano, Investigation, Methodology; Alessandro Brombin, Visualization, Writing—review and editing; Cameron Wyatt, Formal analysis, Validation, Investigation; Witold Rybski, Resources, Maintaining genetic lines and genetic crosses; Leonie F A Huitema, Resources, Generation of transgenic line; Rodney M Dale, Conceptualization, Resources, Writing—review and editing, Result interpretation, and sharing of reagents and expertise; Koichi Kawakami, Christoph Englert, Resources, Writing—review and editing, Result interpretation, and sharing of reagents and expertise; Tamir Chandra, Conceptualization, Software, Formal analysis, Supervision, Funding acquisition; Stefan Schulte-Merker, Nicholas D Hastie, Conceptualization, Resources, Supervision, Funding acquisition, Writing—original draft, Writing—review and editing; E Elizabeth Patton, Conceptualization, Resources, Data curation, Formal analysis, Supervision, Funding acquisition, Investigation, Methodology, Writing—original draft, Project administration, Writing—review and editing

## Author ORCIDs

Alessandro Brombin http://orcid.org/0000-0001-8262-9248
Witold Rybski http://orcid.org/0000-0002-6025-2918
Rodney M Dale https://orcid.org/0000-0003-4255-4741
Koichi Kawakami http://orcid.org/0000-0001-9993-1435
Christoph Englert http://orcid.org/0000-0002-5931-3189
Stefan Schulte-Merker http://orcid.org/0000-0003-3617-8807
E Elizabeth Patton http://orcid.org/0000-0002-2570-0834

### Ethics

Animal experimentation: All work presented in this study has been performed in accordance with the UK legal requirements for the protection of animals used for experimental or other scientific research under the Animal (Scientific Procedures) Act 1986. All experiments were approved by the University of Edinburgh Ethics Committee, and performed under the Home Office Project License 70/8000 to EEP. Zebrafish welfare and husbandry were closely monitored by the MRC Human Genetics Unit Zebrafish Facility staff.

### Decision letter and Author response

Decision letter https://doi.org/10.7554/eLife.30657.028
Author response https://doi.org/10.7554/eLife.30657.029

## Additional files

### Supplementary files

• Supplementary file 1. (a) Single-cell differential expression list. (b) Gene List Sources. (c) Zebrafish cartilage genes. (d) WT1 gene targets. (e) p53 gene targets. (f) WT1 and p53 shared gene targets
DOI: https://doi.org/10.7554/eLife.30657.023

• Supplementary file 2. List of primers used for qRT-PCR and genotyping.
DOI: https://doi.org/10.7554/eLife.30657.024

• Transparent reporting form
DOI: https://doi.org/10.7554/eLife.30657.025

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
