## [Decision Letter]

Thank you for submitting your article "*Wilms Tumor 1b* defines a wound-specific sheath cell subpopulation associated with notochord repair" for consideration by *eLife*. Your article has been reviewed by three peer reviewers, and the evaluation has been overseen by a Guest Reviewing Editor and Didier Stainier as the Senior Editor. The reviewers have opted to remain anonymous.

The reviewers have discussed the reviews with one another and the Reviewing Editor has drafted this decision to help you prepare a revised submission.

Summary:

All three reviewers agreed the manuscript reports an interesting phenomenon that adds new perspectives beyond recently published work by Garcia et al. and that the paper is well presented. The reviewers also pointed out the somewhat descriptive nature of the study in its present form and the need to strengthen key observations. Particularly, the significance of *wt1b* induction, which is the foundation of the whole paper. There was also interest from all in the formation of ectopic vertebrae, potentially via a cartilage intermediate, which is an intriguing possibility that also needs to be substantiated further. Finally, there were also two specific points raising the need for better characterization of the injury model and the regenerative response.

Essential revisions:

1) The emergence of a *wt1b*+ positive cell population that is otherwise not detectable within the developing or mature notochord is supported solely by the use of *Tg(wt1b:GFP)* transgene, the conclusions of the paper would be strengthened by showing that the transgene faithfully recapitulates endogenous *wt1b* expression.

1a) Are *wt1b* mRNA and/or protein excluded from the uninjured, developing notochord?

1b) Is induction of *wt1b* mRNA restricted to the injury site as shown by the transgenic reporter?

2) The significance of *wt1b* induction was not addressed experimentally. Ideally, the authors should attempt a loss of function using a *wt1b* mutant, or a *wt1b*:Nitroreductase cell ablation approach. Alternatively, a gain of function experiment driving *wt1b* mosaically from the *col2* promoter may prove revealing.

3) As stated by the authors, in the heart *wt1b* expression is linked to EMT. Is this also the case in the injured notochord?

4) The authors state that they do not see vacuolated *wt1b* expressing cells at the site of injury, suggesting there are two genetically distinct sheath populations that are involved in different aspects of repair. Were the *wt1b/col2a1a* reporter fish presented in Figure 2—figure supplement 1 followed over to determine whether *col2a1a* positive / *wt1b* negative cells become vacuolated at the site of injury?

5) Is *wt1b* induced in sheath cells following vacuolated cell ablation (e.g. genetic ablation using SAG214:Gal4 and UAS:NTR)? This would determine whether *wt1b* induction is specific to injuries that breach the notochord sheath or not.

6) Given that multiple cell types contribute to centra development, it is unclear in this report where the cartilage forming cells are derived following injury. It is also unclear whether the cartilage that is rapidly observed after injury is directly giving rise to the centrum that later forms at that site. To address these points, at least partially, the authors should perform whole mount in situ hybridization with sclerotome markers and some of the top cartilage hits from the RNA-seq dataset.

7) The authors state that one extra vertebra forms at the site of injury (albeit smaller than normal). The authors should provide some more details about this phenotype, particularly regarding the variability of extra vertebrae. The damage from the injury could cause a single centrum precursor population to be split into two, resulting in the formation of two small centra and an extra vertebra. If the damage occurs at the natural border between two centra, the split in the precursor population would not occur and there would be no extra vertebra. The data in Figure 4 shows that the extra vertebra is smaller in size, and it also appears in Figure 4 that the vertebra adjacent to the ectopic vertebra is also smaller in size. This seems to suggest that an original centrum population is being split into two smaller populations and would imply a wound healing process rather than de novo generation/regeneration of a precursor population. In addition, they should use Safranin-O staining in sections during supernumerary vertebrae formation to better support their conclusions.

[Editors' note: further revisions were requested prior to acceptance, as described below.]

Thank you for resubmitting your work entitled "*Wilms Tumor 1b* defines a wound-specific sheath cell subpopulation associated with notochord repair" for further consideration at *eLife*. Your revised article has been favorably evaluated by a Guest Reviewing Editor and Didier Stainier as Senior Editor.

The authors were fully responsive to the critiques raised by three reviewers and made substantial improvements to their original manuscript which resulted in a much-improved story. There is however one issue detailed below arising from the revised work that needs to be corrected before this manuscript is accepted for publication. There are also small editorial changes and corrections that need to be made. These corrections require in principle only editorial changes.

Main correction:

1) Nystatin cannot be considered a caveolin inhibitor or an inhibitor of caveolae-mediated endocytosis, even if this erroneous assumption is perpetuated in some journals. This does not preclude that nystatin may cause vacuolated cell collapse but cannot be attributed to the inhibition of caveolae. Provided the authors are confident the sheath is not breached under their experimental conditions, the text could be revised to introduce corrections along the lines shown below. Alternatively, the authors may choose to use EM to determine whether treatment with nystatin indeed causes loss of caveolae or assay a function that is unequivocally caveolae dependent.

Text revision:

Subsection title: should be changed to remove the definition of nystatin as a caveolin inhibitor, e.g. "Vacuolated cell collapse leads to an increase in *wt1b:gfp* expression", "Nystatin mediated disruption of vacuolated cells leads to an increase in *wt1b:gfp* expression", or something to that effect.

Text: the references to nystatin as a caveolin or caveolae inhibitor should be removed. Instead the authors may report the (specific?) effect of nystatin in vacuolated cells and mention its "similarity" to the phenotype of caveolin mutants. Using the term phenocopy would imply that other aspects of the phenotype are also recapitulated, which was not examined in this work. As long as the sheath is not breached the conclusions would remain the same.

2) The reference to the manuscript by Lleras et al. should be expressed as personal communication instead of "submitted" at this point. This might be updated later depending on the timing of publication.

Additional points:

Subsection “Wound specific expression of *wt1b* in the notochord”, last paragraph: It would be better to say: "to induce localized damage" instead of "to specifically induce localized damage". The word "specifically" would not be appropriate for this type of injury.

Subsection “*wt1b* expressing cells emerge from the notochord sheath”, first paragraph: *Tg(SAGFF214A:gfp)* labels the cytoplasm of vacuolated cells, not the membrane. Please correct.

Is the signal for the *wt1* antibody lost in the *wt1b* mutant? Please clarify in the text if appropriate and include as much information as possible about this original reagent.

---

## [Author Response]

Summary:All three reviewers agreed the manuscript reports an interesting phenomenon that adds new perspectives beyond recently published work by Garcia et al. and that the paper is well presented. The reviewers also pointed out the somewhat descriptive nature of the study in its present form and the need to strengthen key observations. Particularly, the significance of wt1b induction, which is the foundation of the whole paper. There was also interest from all in the formation of ectopic vertebrae, potentially via a cartilage intermediate, which is an intriguing possibility that also needs to be substantiated further. Finally, there were also two specific points raising the need for better characterization of the injury model and the regenerative response.

We now add substantial new data to our manuscript to address all seven essential revisions. To summarize the changes we have made, we would like to highlight the following major new revisions to our work:

1) Significance of the *wt1b* induction. We have now proved Wt1b protein and *wt1b* gene expression specifically at the site of injury in the notochord, and we have generated and characterised a *wt1b* mutant line. (Essential revisions 1, 2).

2) Further substantiation of the formation of the *wt1b*-expressingvertebrae. We present new single-cell and 10-cell sequencing studies in the zebrafish notochord, and together with validated gene expression array data comparing injured and control regions, provide strong evidence that the *wt1b*+ sheath cells express cartilage and vacuolar cellular pathways, and represses a novel Wt1b-p53 transcriptional programme. These *wt1b*-expressing cells are maintained within the injured vertebrae until adulthood. (Essential revisions 3, 4, 6).

3) Better characterization of the injury model and regenerative response. We have now better characterized the injury model by applying a Caveolin inhibitor to phenocopy the *caveolin* zebrafish mutants and reveal *wt1b+* subpopulations to emerge with and without breach of the sheath. Furthermore, and following an insightful suggestion from one reviewer, we have found that the site of injury does impact upon the final vertebrae number and suggest that the appearance of *wt1b:gfp* cells that stay closely associated with the adult vertebrae at the site of damage reflect a wound healing process. (Essential revisions 5, 7).

Essential revisions:1) The emergence of a wt1b+ positive cell population that is otherwise not detectable within the developing or mature notochord is supported solely by the use of Tg(wt1b:GFP) transgene, the conclusions of the paper would be strengthened by showing that the transgene faithfully recapitulates endogenous wt1b expression.1a) Are wt1b mRNA and/or protein excluded from the uninjured, developing notochord?1b) Is induction of wt1b mRNA restricted to the injury site as shown by the transgenic reporter?

Thank you for this critical comment to address the endogenous expression of *wt1b* in the zebrafish notochord. We do already present evidence of endogenous Wt1b protein expression in Figure 1. We have generated our own antibody toward zebrafish Wt1 and show that it is expressed at the damage site (Author response image 1).

Building on this, we now include immunohistochemistry of notochord sections stained with anti-GFP and anti-WT1 antibodies in injured and non-injured fish (new Figure 1—figure supplement 2). This demonstrates that GFP and Wt1b protein is restricted to the injury site in the notochord, and excluded from the uninjured, developing notochord.

Further, we now provide evidence of endogenous *wt1b* mRNA expression in *wt1b:gfp* -positive cells.As described in more detail below, we have now added new data of single-cell and 10-cell sequencing of FACS sorted cells from the injured *Tg(wt1b:gfp; col2a1a:mCherry)* zebrafish. We find *wt1b* is expressed only in cells that express *gfp* or both *gfp* and *mCherry* but find no *wt1b* expression in the cells that express *mCherry* alone (new Figure 4).

Our antibody and RNA sequencing of sorted cells proves that the transgene faithfully recapitulates endogenous *wt1b* expression in the notochord. Our work proves that *wt1b* expression marks a unique subpopulation in notochord sheath cells following injury in zebrafish.

2) The significance of wt1b induction was not addressed experimentally. Ideally, the authors should attempt a loss of function using a wt1b mutant, or a wt1b:Nitroreductase cell ablation approach. Alternatively, a gain of function experiment driving wt1b mosaically from the col2 promoter may prove revealing.

Thank you for this comment regarding the function of Wt1b in this process. We had started to generate a mutant line, with the intent to address the notochord effects in a *wt1b* mutant situation. Using CRISPR/Cas9 genome editing, we have generated a 5 bp deletion mutant line (new Figure 8—figure supplement 1). At the time of submission, we had not had time to characterise this line. We have confirmed that only the *wt1∆5* mRNA is expressed in mutant embryos (new Figure 8—figure supplement 1). We now have adult homozygous *wt1^∆5/∆5^*mutant fish and have analysed these for *wt1b:gfp* expression, cellularity at the damage site, cartilage formation at the site of injury and in the *wt1b:GFP* expressing adult vertebrae (new Figure 8—figure supplement 1). We do not see an overt phenotype in the *wt1^∆5/∆5^*mutant. However, our new data from the single-cell and 10 cell-sequencing reveals a Wt1 transcriptional signature, coupled with the continued expression of *wt1b:gfp* up to adult stages. This suggests that Wt1 may indeed be more than a marker of sheath cell populations in a wounding response. It is possible that compensatory mechanisms exist, and indeed we find a small but significant increase in *wt1a* expression in injured tissue from the *wt1b^∆5/∆5^* mutants. We now include a section on this in the Discussion.

3) As stated by the authors, in the heart wt1b expression is linked to EMT. Is this also the case in the injured notochord?4) The authors state that they do not see vacuolated wt1b expressing cells at the site of injury, suggesting there are two genetically distinct sheath populations that are involved in different aspects of repair. Were the wt1b/col2a1a reporter fish presented in Figure 2—figure supplement 1 followed over to determine whether col2a1a positive / wt1b negative cells become vacuolated at the site of injury?

Thank you for these important questions. We address reviewer comments 3 and 4 together for ease of discussion.

We are also very excited to understand the nature of the *wt1b:gfp; col2a1a:mCherry* cells. To address this in an unbiased fashion, we have now performed single and 10-cell sequencing of FACS sorted cells from *Tg(wt1b:gfp; col2a1a:mCherry)* zebrafish 3 days following injury.

This is now presented in a new Figure 4. The key findings from this experiment are:

1) The *gfp (wt1b+), mCherry (col2a1a+)* and *gfp,mCherry (wt1b+, col2a1a+)* cells independently cluster into three distinct cell subpopulations of the injured notochord (new Figure 4). This is strong evidence that the transgenic makers reveal true cellular subpopulations in vivo.

2) We do not find evidence of EMT at this stage, but rather observe that the *wt1b+col2a1a+* cell cluster expresses genes that are enriched for genes associated with lysosomes and vacuoles (new Figure 4), and cartilage genes (new Figure 4). This suggests that some of these cells may become vacuolated cells, and we find evidence of this when imaging larvae following injury at a later stage (new Figure 4).

3) The *wt1b+ col2a1a+* cell cluster is significantly associated with repression of Wt1 regulated genes (new Figure 4). Wt1 can function as a transcriptional activator or repressor, and our evidence suggests that Wt1b may function as a repressor for a gene subset in *wt1b+ col2a1a+* cells.

4) Wt1 function is regulated by interactions with transcription co-factors that enable activation or repression of target genes. We screened the *wt1b+ col2a1a+* cell cluster for selective enrichment of Wt1 co-factors and found *tp53* expression specifically enriched in the *wt1b+ col2a1a+* cell cluster (new Figure 4). Remarkably, a p53 gene set enrichment analysis (GSEA) reveals that p53 genes are significantly enriched in the *wt1b+ col2a1a+* cell cluster (new Figure 4). However, genes co-regulated by Wt1 and p53 are selectively repressed in the *wt1b+ col2a1a+* cell cluster (new Figure 4). This was completely unexpected and reveals a novel and unknown Wt1-p53 axis that may function together in the notochord wound process.

5) Is wt1b induced in sheath cells following vacuolated cell ablation (e.g. genetic ablation using SAG214:Gal4 and UAS:NTR)? This would determine whether wt1b induction is specific to injuries that breach the notochord sheath or not.

We appreciate that the *Tg(SAG214A:Gal4; USA:NTR)* approach would be informative, however, we do not have the combination of these transgenic lines in house, and to obtain them would have required to cross two different transgenes onto the *Tg(wt1b:gfp)* transgenic background (two generation times plus shipping of animals between different sites).

Instead, we have tried to phenocopy the *cav1* mutant phenotypes (Garcia et al., 2017). In our first attempts, we used a morpholino and a crispant approach to target *cav1*, but we were unable to see the altered cellularity at the notochord site.

As an alternative approach, we used a small molecule caveolin inhibitor called Nystatin (Figure 2—figure supplement 2). Nystatin binds sterols and inhibits caveolae-mediated endocytosis. Remarkably, treatment of *Tg(wt1b:gfp; col2a1a:mCherry)* zebrafish with nystatin showed a dose-dependent increase in cellularity in the notochord, similar to *cav1, cav3* and *cavin1b* mutants. GFP was strongly expressed in a subpopulation of the mCherry-positive sheath cells (white arrows), and cells expressing both *gfp* and *mCherry* were clearly visible at the site of cellularity (yellow arrows).

Our work now shows that *wt1b* expression defines the subpopulation of sheath cells that fill gaps in the vacuolated notochord following the breach of the notochord sheath and following notochord vacuolar dysfunction. We are excited that our work is applicable to the conditions of vacuolated cell damage and have included this data in the Results and Discussion sections.

6) Given that multiple cell types contribute to centra development, it is unclear in this report where the cartilage forming cells are derived following injury. It is also unclear whether the cartilage that is rapidly observed after injury is directly giving rise to the centrum that later forms at that site. To address these points, at least partially, the authors should perform whole mount in situ hybridization with sclerotome markers and some of the top cartilage hits from the RNA-seq dataset.

We agree that this is an interesting question, but since the (chorda)centrum is an acellular structure, it is not possible to show which cells contribute to it, at least during the first steps of mineralisation and before any osteocytes appear (which is much later). However, since *entpd5* is essential for mineralization and since we see *entpd5a*+ notochord sheath cells in the wound area, it seems likely that sheath cells, in conjunction with cartilage formation at the site of injury play a critical role in centrum formation. We have extended the Discussion accordingly to discuss this point in more detail.

In further response to this question, we have isolated sections of the damage site and control undamaged tissue and performed qRT-PCR for the *matrix gla protein* (*mgp)* and *Sox9* mRNA expression. We chose these two genes because *mgp* was highly expressed in the microarray analysis and important for bone organisation, and because Sox9b is a master cartilage transcription factor. Our new findings demonstrate that *mgp* RNA is increased in notochord damaged tissue, compared with undamaged notochord tissue (Figure 3). This validates the microarray analysis and supports the wound specific upregulation of cartilage genes.

To address which cells in the wound response express the cartilage genes, we have isolated FACS sorted cells from Tg(*wt1b:gfp; col2a1a:mCherry)* injured fish and find *sox9b* and *mgp* to be selectively upregulated in the GFP^+^mCherry^+^ cell populations(Figure 4).These findings indicate that the *wt1b*-positive sheath cells express the cartilage genes and may co-ordinate the cartilage response at the wound site.

7) The authors state that one extra vertebra forms at the site of injury (albeit smaller than normal). The authors should provide some more details about this phenotype, particularly regarding the variability of extra vertebrae. The damage from the injury could cause a single centrum precursor population to be split into two, resulting in the formation of two small centra and an extra vertebra. If the damage occurs at the natural border between two centra, the split in the precursor population would not occur and there would be no extra vertebra. The data in Figure 4 shows that the extra vertebra is smaller in size, and it also appears in Figure 4 that the vertebra adjacent to the ectopic vertebra is also smaller in size. This seems to suggest that an original centrum population is being split into two smaller populations and would imply a wound healing process rather than de novo generation/regeneration of a precursor population. In addition, they should use Safranin-O staining in sections during supernumerary vertebrae formation to better support their conclusions.

We thank the reviewers for their insightful question regarding the possibility that we are damaging a precursor population upon injury. At 3 dpf, we do not have a means to visualise a centrum precursor population, however, by day 5 and 7 pf we can visualize the patterned expression of *entpd5a* in the notochord using Tg(*entpd5a:kaede*). New evidence demonstrates that metameric expression of *entpd5a* in notochord sheath cells is an essentialrequirement for the patterned formation of chordacentra rings (Lleras Forero et al., submitted). Indeed, the reviewer is correct, and we find that the extra vertebrae is present only upon damaging an *entpd5a* expression domain (new Figure 6—figure supplement 1). This indicates that the wound response alone is not sufficient to induce an extra vertebra, and that *entpd5a* cell population domains are likely to be important in the vertebra formation at the wound site.

Nonetheless, we consistently find that the vertebra(e) that forms at the site of the damage is/are significantly smaller in a given space interval and have quantified this (Figure 5). Further, we find it remarkable that the *wt1b:gfp* response is long-lasting over time, and that *wt1b:gfp* expressing cells remain associated with the vertebra(e) at the damage site. To address this phenomenon, we have been able to improve our confocal imaging of Tg(*wt1b:gfp*) and alizarin staining in live adult fish (new Figure 7) and continue to observe *wt1b:gfp* expressing cells associated with the single vertebra or two vertebrae at the site of damage until adulthood, and (at least one of) these vertebrae are smaller. We have now amended the text and Figure 7 (previous Figure 6) to incorporate this change and emphasise what we believe are the first steps to understanding this important wound-healing process.

[Editors' note: further revisions were requested prior to acceptance, as described below.]

The authors were fully responsive to the critiques raised by three reviewers and made substantial improvements to their original manuscript which resulted in a much-improved story. There is however one issue detailed below arising from the revised work that needs to be corrected before this manuscript is accepted for publication. There are also small editorial changes and corrections that need to be made. These corrections require in principle only editorial changes.Main correction:1) Nystatin cannot be considered a caveolin inhibitor or an inhibitor of caveolae-mediated endocytosis, even if this erroneous assumption is perpetuated in some journals. This does not preclude that nystatin may cause vacuolated cell collapse but cannot be attributed to the inhibition of caveolae. Provided the authors are confident the sheath is not breached under their experimental conditions, the text could be revised to introduce corrections along the lines shown below. Alternatively, the authors may choose to use EM to determine whether treatment with nystatin indeed causes loss of caveolae or assay a function that is unequivocally caveolae dependent.Text revision:Subsection title: should be changed to remove the definition of nystatin as a caveolin inhibitor, e.g. "Vacuolated cell collapse leads to an increase in wt1b:gfp expression", "Nystatin mediated disruption of vacuolated cells leads to an increase in wt1b:gfp expression", or something to that effect.

Thank you, as suggested, this has now been changed to: “Nystatin mediated disruption of vacuolated cells leads to an increase in *wt1b:gfp* expression”.

Text: the references to nystatin as a caveolin or caveolae inhibitor should be removed. Instead the authors may report the (specific?) effect of nystatin in vacuolated cells and mention its "similarity" to the phenotype of caveolin mutants. Using the term phenocopy would imply that other aspects of the phenotype are also recapitulated, which was not examined in this work. As long as the sheath is not breached the conclusions would remain the same.

Thank you, this text now reads:

“Nystatin mediated disruption of vacuolated cells leads to an increase in *wt1b:gfp* expression

We tested if the *wt1b*-response was specific to wounds that involved rupture of the sheath, or if *wt1b* expressing cells could be induced upon loss of vacuolated cell integrity alone. […] Thus, expression of *wt1b* in the sheath does notrequire a physical breach of the sheath, and *wt1b* expression may be applicable to a wider range of tissue stress and damage situations.”

We have also changed the heading of Figure 2—figure supplement 2 to read:

“Figure 2—figure supplement 2. Nystatin treatment leads to upregulation of *wt1b:gfp* expression in notochord sheath cells”.

2) The reference to the manuscript by Lleras et al. should be expressed as personal communication instead of "submitted" at this point. This might be updated later depending on the timing of publication.

Thank you, this has now been changed to LL-F and SS-M, personal communication.

Additional points:Subsection “Wound specific expression of wt1b in the notochord”, last paragraph: It would be better to say: "to induce localized damage" instead of "to specifically induce localized damage". The word "specifically" would not be appropriate for this type of injury.

Thank you, this has now been changed.

Subsection “wt1b expressing cells emerge from the notochord sheath”, first paragraph: Tg(SAGFF214A:gfp) labels the cytoplasm of vacuolated cells, not the membrane. Please correct.

Thank you, this is now corrected.

Is the signal for the wt1 antibody lost in the wt1b mutant? Please clarify in the text if appropriate and include as much information as possible about this original reagent.

We have not tested the antibody on the Wt1b mutants, however please note that the antibody is raised against an epitope that is present in both Wt1a and Wt1b, so complete loss of Wt1 signal may only be detectable in a wt1a^-^wt1b^-^ double mutant, which is not available at this time.

Please find a new paragraph that has been added to the Materials and methods section with more detail about the Wt1 antibody. We tested the antibody by western blot and see a very strong band at 43kDa, consistent with the molecular weight of Wt1 proteins in zebrafish. We used a peptide competition assay by immunofluorescence to demonstrate that the antibody is highly selective for the Wt1 epitope.

“αWt1 zebrafish antibody

The Wt1 antibody was synthesised by Cambridge Research Biochemicals (CRB) antibody production services (http://www.crbdiscovery.com/home). […] Immunofluorescence on paraffin-embedded sections with Wt1 antibody (diluted 1:33,000) revealed cell specific staining in the kidney and notochord wound site that was depleted by co-incubation of the Wt1 antibody with the Wt1 epitope peptide.”